# W-CTC: a Connectionist Temporal Classification Loss with Wild Cards

**Xingyu Cai, Jiahong Yuan, Yuchen Bian, Guangxu Xun, Jiaji Huang, Kenneth Church**
Baidu Research, 1195 Bordeaux Dr, Sunnyvale, CA 94089, USA
xingyucai@baidu.com

## Abstract

Connectionist Temporal Classification (CTC) loss is commonly used in sequence learning applications. For example, in Automatic Speech Recognition (ASR) task, the training data consists of pairs of audio (input sequence) and text (output label), without temporal alignment information. Standard CTC computes a loss by aggregating over all possible alignment paths, that map the entire sequence to the entire label (full alignment). However, in practice, there are often cases where the label is incomplete. Specifically, we solve the partial alignment problem where the label only matches a middle part of the sequence. This paper proposes the wild-card CTC (W-CTC) to address this issue, by padding wild-cards at both ends of the labels. Consequently, the proposed W-CTC improves the standard CTC via aggregating over even more alignment paths. Evaluations on a number of tasks in speech and vision domains, show that the proposed W-CTC consistently outperforms the standard CTC by a large margin when label is incomplete. The effectiveness of the proposed method is further confirmed in an ablation study.

## 1 Introduction

Connectionist Temporal Classification (CTC) was proposed in (Graves et al., 2006) to train end-to-end sequence learning models. It has become the most popular loss function in many sequence learning applications, such as Automatic Speech Recognition (ASR) (Amodei et al., 2016; Battenberg et al., 2017) and Optical Character Recognition (OCR) (Liu et al., 2015; Subramani et al., 2020; Chen et al., 2021). CTC has a number of attractive properties. For example, it comes with a clear interpretation that maximizes the posterior marginal probability summed over all possible alignment paths. In addition, CTC loss can be easily incorporated into neural nets because it is fully differentiable. The gradients can be estimated efficiently by borrowing the well-known forward-backward algorithm from Hidden Markov Models (HMMs) (Rabiner, 1989).

One limitation of standard CTC is that it assumes the label matches the whole input sequence. CTC performs a full alignment between them. However, we often face the partial alignment in practice. For example, a speech transcription often omits the prefacing introduction of the speaker; a beginning of an audio book that tells who read the book, along with various other details, are typically not in the written version; a sign language interpreter often ignore the irrelevant gestures in the beginning and ending stages; etc. The above motivating examples are also referred as the streaming problem (Sakurai et al., 2007). In a streaming speech or video, we only capture a fraction of interested segments to form the labels. It is also common that the label is corrupted or noisy due to variable recording conditions. Partial alignment is useful in semi-supervised or transfer learning as well, which enables us to use a coarse model (from a different domain) to automatically generate labels. We can discard the low-confident outputs and form the incomplete, but more accurate labels, then feed them into the target model to train. In time series domain, streaming Dynamic Time Warping (s-DTW) was proposed (Sakurai et al., 2007) for partial warping, and Cai et al. (2019) extended it to differentiable training in neural nets. Inspired by their work, we propose the wild-card enhanced CTC, dubbed as W-CTC, to perform the end-to-end partial-alignment sequence learning. To the best of the authors' knowledge, this is the first study to apply CTC on incomplete labels.

### 1.1 Problem Formulation

Let $X = [X_0, X_1, \ldots, X_{T-1}]$ be the input sequence of length $T$, e.g., speech audio frames for an ASR system, or image segments for an OCR system. Let $Y = [Y_0, Y_1, \ldots, Y_{N-1}]$ be the corresponding targets (labels) of length $N$, e.g., the transcription for ASR, or the recognized string for

OCR. Assume $T \geq N$. Though both $X$ and $Y$ are provided, we don't know how they are aligned. Let the alignment function be $\psi : X \to Y$, which maps a frame $x \in X$ to a target $y \in Y$. Define:

**Definition 1** (**Subsequence**). *A **subsequence** of $X$, denoted as $\hat{X} \subset X$, is a vector of multiple **consecutive** elements in $X$, i.e. $\hat{X} = [X_i, X_{i+1}, \ldots, X_{i+\tau}]$, where $\tau > 0$.*

**Definition 2** (**Full alignment**). *A **full** alignment maps the entire $X$ to the entire $Y$, i.e. $y = \psi(x) \in Y$, for any $x \in X$, as well as $x = \psi^{-1}(y) \in X$, for any $y \in Y$.*

**Definition 3** (**Alignment path**). *An alignment **path**, $\pi$, is a vector of the same length as $X$, i.e. $\pi = [y_0, y_1, \ldots, y_{T-1}]$, where each element $y_i = \psi(X_i) \in Y$, for $0 \leq i < T$.*

**Definition 4** (**Valid alignment**). *An alignment path is **valid**, if it satisfies:*

- *Monotonicity: if $X_i$ maps to $Y_j$, $X_{i+t}$ cannot map to $Y_{j-k}$, for any $t > 0$ and $k > 0$.*
- *Surjection (many-to-one mapping): $\psi(\cdot)$ is a surjective function, such that a subsequence $\hat{X} \subset X$, could maps to a single $Y_j \in Y$. But each $X_i$ can have only one label $y \in Y$.*

Definition 4 is a common assumption, because we often have a higher sampling rate in audios or images, for a better resolution. In consequence, $T \geq N$, thereafter multiple frames correspond to a single target. In practice, $X$ could be the sequence of frame embedding vectors, i.e. $X_i \in \mathcal{R}^d$, obtained by a neural network. Provided the targets $Y$, we want to train the neural network by implicitly learning the alignment between $X$ and $Y$. This is a **full alignment problem** and often handled by CTC loss (Graves et al., 2006) or attention mechanism (Bahdanau et al., 2014).

In this paper, we extend the full alignment problem, to a more general partial alignment case, where the input could be incomplete. That is, the input sequence $X$, does not necessarily have an end-to-end mapping to the label $Y$ (violate Definition 2). In particular, we solve the missing $Y$ problem, where $X$ is complete. For example, the annotated transcription is incomplete, or corrupted, such that it does not correspond to the full audio speech. It is sometimes referred as the streaming alignment problem, as $X$ can be seen as a long stream signal, while $Y$ only matches part of $X$.

**Definition 5** (**Partial Alignment Problem**). *A problem where the label is a subsequence $\hat{Y} = [Y_k, Y_{k+1}, \ldots, Y_{k+L-1}] \subset Y$, that has a length $L < N$. Both $k$ and $L$ are **unknown**.*

Note that the missing portion are at the beginning and the end of $Y$. A commonly seen example in ASR system, is the silence/noise at the two sides of $X$. The transcription often does not include such silence/noise remarks, resulting in a mismatch at the two ends. This is already handled by CTC loss with an additional "blank" label (refer to (Graves et al., 2006)). We tackle the more complex problem, that the missing part is meaningful rather than just noise/silence.

## 1.2 Brief Introduction of CTC

To introduce CTC, let's first consider the **full alignment problem**. There are many paths leading to the same label $Y$ from input $X$. Let $\mathcal{Z}$ be the set of all valid paths. The standard CTC loss (Graves et al., 2006) is defined as the negative log-likelihood of the labels given input sequences, i.e.

$$\mathcal{L}_{\text{CTC}}(X, Y) = -\log P(Y|X) = -\log \sum_{\pi \in \mathcal{Z}} P_\pi(Y|X) \tag{1}$$

where $P_\pi(Y|X)$ is the probability of a particular alignment path $\pi$. For any path, CTC assumes conditional independence amongst $X_i$'s, $0 \leq i < T$. So the probability of that path, is approximated by a product of the path-determined label's probability at each frame, i.e. $P_\pi(Y|X) \approx \prod_{0 \leq t < T} P_t(\pi[t] \mid X)$. Minimizing the CTC loss is equivalent to maximizing the likelihood of the observed label given the input sequence. CTC is fully differentiable, making it favorable to train an ASR or OCR system, where no exact temporal alignment information is provided. CTC is calculated using Dynamic Programming (DP), which will be further analyzed in Section 2.1.

Our target in this paper is the **partial alignment problem**, which could reduce to the full alignment problem, by trying to do full alignment between the corrupted targets, $\hat{Y}$, and all possible subsequences of $X$. Let $\hat{X}_{i,\tau} = [X_i, X_{i+1}, \ldots, X_{i+\tau}]$ be one subsequence, starting from $i$-th frame with a length of $\tau + 1$. The straight-forward way is to loop over all possible $(i, \tau)$ pairs, obtain the CTC loss for each subsequence, and summarize them, i.e.,

$$\mathcal{L}'_{\text{CTC}}(X, \hat{Y}) = \sigma\{\mathcal{L}_{\text{CTC}}(\hat{X}_{i,\tau}, \hat{Y}) \mid i + \tau < T\} \tag{2}$$

where $\sigma\{\cdot\}$ means a summary of the set, e.g., summation or max.

This straight-forward approach is too expensive to compute: For each $\mathcal{L}_{\text{CTC}}(\hat{X}_{i,\tau}, \hat{Y})$, a DP is utilized, leading to $O(TN)$ cost. Unfortunately, DP cannot be completely parallelized. Looping through all $(i, \tau)$ pairs yields a total complexity of $O(T^3N)$. A trick is that when $i$ is fixed, we can reuse the DP trellis from $\mathcal{L}_{\text{CTC}}(\hat{X}_{i,\tau}, \hat{Y})$ to obtain $\mathcal{L}_{\text{CTC}}(\hat{X}_{i,\tau+1}, \hat{Y})$, at a cost of $O(N)$ instead of $O(TN)$. Even with this trick, $O(T^2N)$ is still required as looping over $i$ cannot be avoided. Considering $T$ can be fairly large for audio frames, this approach is hardly deployable in practice.

# 2 PROPOSED WILD-CARD CTC

## 2.1 REVIEW OF CTC COMPUTATION

Before presenting W-CTC, we briefly review in detail the Dynamic Programming (DP) based forward algorithm to compute the CTC loss in Equation 1. Taking an English ASR system as an example, let $X$ be a sequence of embedding vectors of $T$ speech frames, and $Y$ be the corresponding transcription of length $N$. Since there could be silence frames in $X$, which does not correspond to any English letter, the first step is to augment $Y$ by inserting a "blank" label that could represent the silence. "blank" is inserted between any two characters, as well as the beginning and the end of the transcription. For instance, a transcription of "hello" will be augmented to "-h-e-l-l-o-", where "-" represents the "blank" label. The augmented transcription $\tilde{Y}$ has a length $2N + 1$.

The second step is to build the trellis (DP matrix) $M$, which has $2N + 1$ rows and $T$ columns, as illustrated in Figure 1a. Each node, $M_{i,j}$, denotes a subproblem's likelihood, i.e. $M_{i,j} = P(\tilde{Y}[0 : i] \mid X[0 : j])$. Since the speech signal could end with the transcription's last English letter, or with additional silence, the final CTC loss is a combination of these two cases. More specifically, it is the negative log of the sum of the two final states, i.e. $\mathcal{L}_{\text{CTC}}(X, \tilde{Y}) = -\log(M_{2N-1,T-1} + M_{2N,T-1})$. These two final-state nodes are colored orange at the right-bottom corner of Figure 1a.

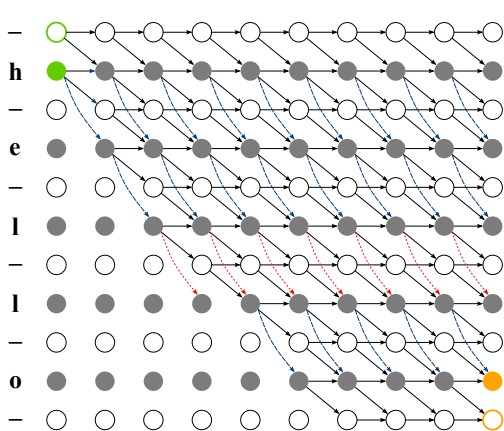
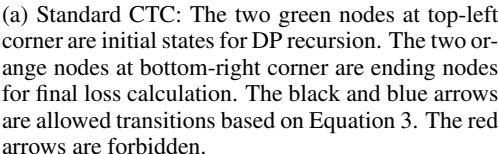
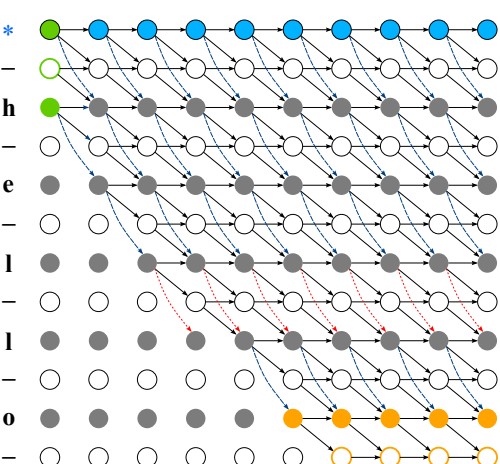

(a) Standard CTC: The two green nodes at top-left corner are initial states for DP recursion. The two orange nodes at bottom-right corner are ending nodes for final loss calculation. The black and blue arrows are allowed transitions based on Equation 3. The red arrows are forbidden.

(b) W-CTC: The first row corresponds to the prepended wild-card "∗" symbol (the blue nodes). There are three initial states. The ending nodes are the entire last two rows, rather than only two right-bottom nodes. These changes enable $\hat{Y}$ to match only a fraction of $X$.

Figure 1: Illustration of Dynamic Programming based CTC and proposed W-CTC loss calculation.

To fill the trellis, we start from the top-left corner. The value of $M_{0,0}$ is the probability of "blank" given the first frame, i.e. $M_{0,0} = P(-|X_0)$. Similarly, $M_{1,0} = P(Y_0|X_0)$ is the probability of transcription's first letter. The speech could begin with either silence or the transcription's first English letter, so both $M_{0,0}$ and $M_{1,0}$ are valid initial states (the green nodes at top-left corner in Figure 1a). Note that the probability of a predicted symbol, i.e. $P(\tilde{Y}_i|X_{0:j})$, is often obtained by projecting the embedding vector $X_j$ to a logit vector (length is the augmented vocabulary size), then

applying softmax operation. As the prediction at any time step can be conditioned on the entire sequence of $X$ (e.g., using a bi-directional LSTM model), we write $P(\tilde{Y}_i|X)$ instead of $P(\tilde{Y}_i|X_{0:j})$ to account for the more general cases. Referring to the illustration in fig. 1a, the rules for updating the remaining values $M_{i,j}$ are as follows:

$$M_{i,j} = \begin{cases} P(\tilde{Y}_i|X) \cdot (M_{i,j-1} + M_{i-1,j-1}), & \text{if } \tilde{Y}_i = \text{``-''} \text{ or } \tilde{Y}_i = \tilde{Y}_{i-2} \\ P(\tilde{Y}_i|X) \cdot (M_{i,j-1} + M_{i-1,j-1} + M_{i-2,j-1}), & \text{otherwise} \end{cases} \tag{3}$$

The line 1 in Equation 3 does not have the term $M_{i-2,j-1}$, which means: (1) If $\tilde{Y}_i$ is "blank", the path cannot come from a skip "blank" before ($M_{i-2}$). (2) If two consecutive letters are the same, e.g., "-l-l-" in "-h-e-l-l-o-", the node corresponding to the second "l" cannot directly have a path from the node corresponding to the first "l". In this case, the "blank" symbol between two "l"s plays a separator role, enabling the decoder to know there are two "l"s instead of one. As a result, the red arrows in Figure 1a are forbidden. In other cases, the node $M_{i,j}$ could have a path from $M_{i-2,j}$, so the blue arrows are allowed in Figure 1a.

Following above DP recursion rules, the trellis $M$ is filled and the values of the two ending nodes (orange nodes at bottom-right corner), are added to obtain the final CTC loss. To achieve numerical stability, the computation is typically done in log scale. This is the forward calculation step. The backward calculation is similar and we omit it due to space limit. Interested readers can refer to (Graves et al., 2006) for details.

## 2.2 THE PROPOSED W-CTC ALGORITHM

The label $Y$ could be corrupted, e.g., the speech utterance still corresponds to "hello", but the annotated label is only "ell". The original CTC does not work here because it assumes a full alignment between the speech and the transcription (both have to be complete). Inspired by (Sakurai et al., 2007) and (Cai et al., 2019), we modify the original CTC loss computation procedure as follows:

- Not only inserting a "blank" symbol between characters, we also prepend the augmented transcription with a wild-card symbol "$*$". For example, "hello" is converted to "$*$-h-e-l-l-o-". The wild-card symbol could represent any character in vocabulary including the "blank" symbol, therefore perfectly matches any speech frames. In other words, $P(*|X) = 1$. As a result, the augmented $\tilde{Y}$ now has a length $2N + 2$, and the trellis $M$ has $2N + 2$ rows.

- Instead of two initial-state nodes (the green nodes), we also include the first node that corresponds to the wild-card "$*$", leading to a total of 3 starting nodes. This is illustrated in Figure 1b. The top-left 3 green nodes are valid initial states, where $M_{0,0} = P(*|X_0) = 1$.

- Instead of only considering two final states $M_{2N-1,T-1}, M_{2N,T-1}$, we consider the entire last two rows (the bottom two rows are all colored orange in Figure 1b). That is, we collect a set of $\mathcal{L}_{\text{CTC}}^{(j)} = -\log(M_{2N-1,j} + M_{2N,j})$, for $j = N - 1$ to $T - 1$, then we summarize them to obtain our W-CTC loss, i.e. $\mathcal{L}_{\text{W-CTC}} = \sigma\{\mathcal{L}_{\text{CTC}}^{(j)} \mid N - 1 \leq j < T\}$. We can safely ignore the case of $j < N - 1$ as those are not feasible alignments (the probability $P(Y|X[0:j]) = 0$ if $j < N - 1$), because at least $N$ frames are needed to match $N$ characters in the transcription. So the left nodes in the last two rows in Figure 1b, are not colored orange (not ending nodes). Here the function $\sigma\{\cdot\}$ can be chosen from max, summation, or weighted sum.

The rest of the algorithm is the same as the standard CTC. The DP recursion is the same as Equation 3. Empirically, we found that choosing weighted sum for $\sigma\{\cdot\}$, yields the best results:

$$\mathcal{L}_{\text{W-CTC}} = \sum_{N-1}^{T-1} w_j \mathcal{L}_{\text{CTC}}^{(j)} \text{ , where } [w_{N-1}, \ldots, w_{T-1}] = \text{softmax}\left([-\mathcal{L}_{\text{CTC}}^{(N-1)}, \ldots, -\mathcal{L}_{\text{CTC}}^{(T-1)}]\right) \tag{4}$$

An ablation study on different strategies for $\sigma\{\cdot\}$, is presented in Section 3.4. Note that since an additional wild-card symbol with a constant probability 1 is inserted for each frame, the total probability becomes unnormalized. However, empirically we found that there is no clear difference whether to normalize the total probability. A detailed discussion is provided in Appendix B. Therefore, we use the unnormalized likelihood for simplicity.

**Complexity:** Since we don't introduce additional cost except for adding a row in the trellis, the time complexity remains the same as the original CTC computation, i.e. $O(TN)$. The DP updates are

the same. It can be speed up by computing the whole column in parallel, resulting in a parallel time complexity of $O(T)$ using $O(N)$ cores. The space complexity is $O(TN)$ as we need to store the graph for backpropogation.

## 2.3 THE KEY SUMMARY

Compared to the standard CTC, the key changes are: (1) Prepend a "∗" symbol on $Y$, resulting in an additional row on top of the trellis $M$ (the blue nodes); (2) There are 3 green nodes to start DP recursion; (3) The orange ending nodes (final states) are the entire last two rows.

The 1st change (prepend a wild-card symbol), ensures that the transcription can start anywhere in the speech frames. Due to $P(∗|X) = 1$, there will be no punishment if going from $M_{0,j-1}$ to $M_{0,j}$ (the probability remains the same). The 3rd change (consider last two rows), make the transcription can end anywhere in the frames, rather than matching to the end of the speech. It is equivalent to implicitly adding a wild-card at the end of the transcription. As a result, the explicit front wild-card, and the implicit final wild-card, successfully handle the missing transcription in both ends.

Combining the two changes, we convert the standard CTC built only for **full** alignment, to a very flexible **partial** alignment usage, at negligible cost. The computation is still fully differentiable. Therefore, we can still obtain the gradient and learn the network parameters, even if the transcription only matches the middle part of the speech. These will be validated in the experiment section.

It is worth noting that, the wild-card symbol "∗", is **NOT** equivalent to the "blank" symbol. The "blank" symbol is within the augmented vocabulary, which has a physical meaning of silence, background noise, or an unknown character. The network still needs to predict the probability of a "blank" symbol at each frame, given the speech. Usually, this probability $P(\text{-}|X)$ is $< 1$. On the contrary, the wild-card "∗" is not in the vocabulary. It matches any character in the vocabulary. The probability $P(∗|X)$ is always 1. Only the wild-card symbol can let the alignment skip the beginning speech frames ($M_{0,0} \to M_{0,1} \to M_{0,2} \to \ldots$), without any punishment. Just putting the "blank" symbol in the first row won't be able to achieve this.

**Limitations:** There are a few limitations of the proposed method. (1) It only handles missing transcription (missing $Y$) problem, but not the case where speech frames are missing (missing $X$). Since one key assumption is the many-to-one mapping ($\psi(\cdot)$ is surgective), a simple extension of prepending a wild-card in front of $X$, will not work. However, this can be addressed by recording longer audio/videos to have some redundancy at both sides, which covers all the labels. (2) The missing part in $Y$, needs to be at two sides, but not in the middle. Taking the "hello" example, it cannot solve the corrupted transcription "h???o", where the "?" symbol represents the missing part. But it can handle the "?ell?" case. (3) This approach requires the remaining fraction $\hat{Y}$ be a continuous subsequence of $Y$, e.g., "?ell?" can be solved, but "?e?lo" cannot.

## 3 EXPERIMENTS

We conduct experiments on speech, OCR and Sign Language tasks in §3.1, §3.2, §3.3, respectively. An ablation study is discussed in §3.4. We implement the forward computing scheme and leverage the auto-grad mechanism to achieve backpropogation. All the codes can be found at https://github.com/TideDancer/iclr22-wctc.

**Experimental Settings:** For each dataset, we randomly mask out a portion $r$ of each training label. For example, if $r = 0.5$, all the training labels will have half of their text masked out. The model is expected to learn from the corrupted labels. The same training procedure is used for both the proposed W-CTC as well as the baseline (standard CTC). The **mask-ratio** $r$ ranges from 0 (no corruption) to 0.9 (90 % is missing). After training, we evaluate on the test set. Note that while the labels in the training set are masked, the ground-truth labels in the test set for evaluation, are not corrupted. We perform 3 runs, and plot the averages and error bars (standard deviations).

## 3.1 AUTOMATIC SPEECH RECOGNITION (ASR) AND PHONEME RECOGNITION (PR)

We use the TIMIT (Garofolo, 1993) dataset in this experiment. The TIMIT dataset includes audios and corresponding transcriptions, at word level and phoneme level. All of the materials are in English. Word transcriptions include 26 English letters plus 10 digits, and phoneme transcriptions

use 61 phonemes. There are 630 speakers. Audio is recorded at a sampling rate of 16 kHz. The transcriptions include timestamps specifying which words and phonemes were spoken when.

Our experiments consist of two tasks: (a) Automatic Speech Recognition (ASR) and (b) Phoneme Recognition (PR). The standard word-error-rate (WER) and phoneme-error-rate (PER) are used to evaluate ASR and PR, respectively. Both WER and PER compare a hypothesis sequence to a reference sequence, where the hypothesis is the predicted sequence, and the reference is the gold sequence (the labels). The comparison counts the number of substitutions, $S$, deletions, $D$, and insertions, $I$. Therefore, WER is defined as $\frac{S+D+I}{N}$, where $N$ is the length of the reference (in words). PER is similar to WER but based on phonemes. Note that WER and PER can exceed 100%.

The backbone model is the pretrained Wav2vec-2.0-base (Baevski et al., 2020). We do fine-tuning until convergence. See Appendix A.1 for more details on the training process. See Appendix D.1 for PER results based on a reduced phoneme inventory of 39 phoneme classes (as opposed to 61). These 39 classes are used by a number of other papers, e.g., (Lopes & Perdigao, 2011; Lee & Hon, 1989).

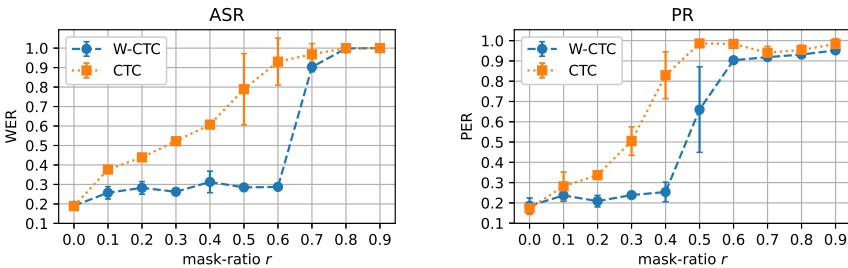

Figure 2: WER / PER vs mask-ratio in ASR and PR tasks, on TIMIT test set.

Figure 2 shows the results on the two tasks: ASR (left panel) and PR (right panel). Both panels plot error rates as a function of mask-ratio $r$. From the figure, we can see: (1) On both tasks, the models trained using W-CTC achieve better results than standard CTC. The blue curves almost always stay lower than the orange curves. (2) The performance of W-CTC is very stable when a small portion of label is masked. For example, when $r = 0.5$ in ASR experiment, the model trained using W-CTC can still get below 30% WER, while the model using standard CTC has 80% WER (almost not usable in practice). (3) The performance of standard CTC deteriorate nearly linearly against mask-ratio, while W-CTC only deteriorates when more than 50% label is missing. (4) When mask-ratio is higher, e.g., $r > 0.7$, though W-CTC is still slightly better, but as expected, both models struggle and yield more than 90% WER. (5) The only exception is that when $r = 0$ (label is completely clean), the model using standard CTC is slightly better than W-CTC. The reason is the wild-card symbol introduced in W-CTC, brings significantly more alignment paths, therefore increases uncertainty and hurt the performance when data is absolutely clean.

## 3.2 Optical Character Recognition (OCR)

The Optical Character Recognition (OCR) task inputs an image containing text, and outputs the recognized text with their locations on the image. OCR systems typically consist of two major components: text detection and text recognition. Text detection creates bounding boxes to segment candidate patches that contain text, and text recognition converts the patches into strings. In the training phase, it cuts the patches (containing text) into "frames", and applies either attention or CTC to do implicit alignment with the text labels. Recent survey papers, e.g., (Chen et al., 2021; Subramani et al., 2020), describe the state-of-the-art OCR systems.

In this paper, we focus on the text recognition block as CTC is widely adopted here. We tested the proposed W-CTC using the well-known CTC-based OCR model, CRNN (Shi et al., 2016). Our implementation is based on (Sun et al., 2020), with a backbone ResNet (He et al., 2016). Testing and training materials were obtained from (Baek et al., 2019). Two standard collections were used for training: (a) MJSynth (MJ, 9 million images) (Jaderberg et al., 2014) and (b) SynthText (ST, 800k images) (Gupta et al., 2016). Each image is $32 \times 100$ pixels. We trained for 150k iterations, with a batch size of 500 images. After training, the evaluations are performed on 7 standard benchmarks: IIIT5K_3000 (Mishra et al., 2012), SVT (Wang et al., 2011), IC03_867 (Lucas et al., 2005), IC13_1015 (Karatzas et al., 2013), IC15_2077 (Karatzas et al., 2015), SVTP (Phan et al., 2013) and

CUTE80 (Risnumawan et al., 2014). The evaluation metric is the recognition accuracy, ACC, defined as the number of correct images divided by total number of images. Images count as correct if and only if the predicted text matches the ground truth exactly. The higher ACC, the better results.

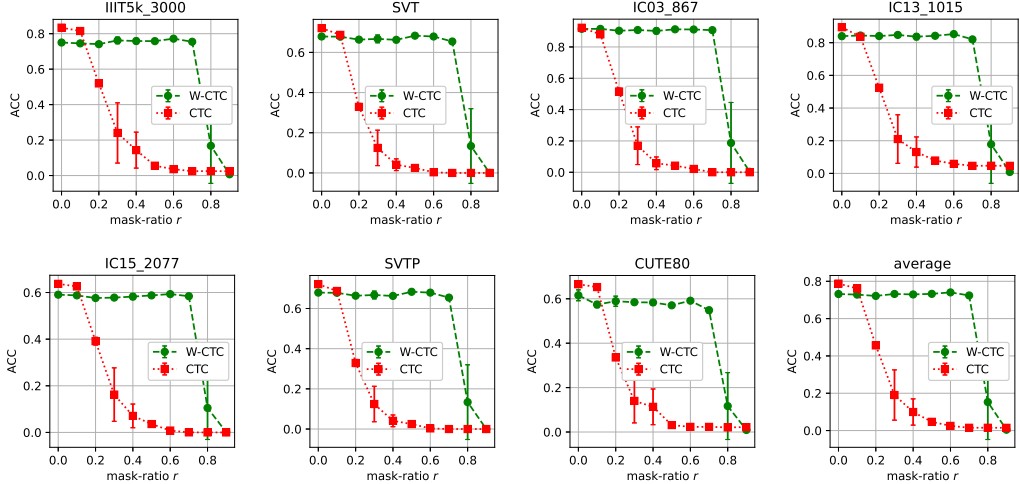

Figure 3: Test accuracy on 7 standard test sets, as a function of $r$ (mask ratio). The last plot is the average. The proposed W-CTC has generally better accuracy than the standard CTC.

**Evaluation Accuracy:** Figure 3 shows a consistent results over different testsets. For standard CTC (red curves), when $r > 0.1$, the performance drops significantly. On the contrary, the model trained using W-CTC (green curves), performs very stable even until 70% of label masked. This indicates that the model can successfully learn the image embeddings, as well as the mappings to characters, from partial labels using W-CTC. Similarly, when data is nearly clean ($r \leq 0.1$), there is a small gap between W-CTC and standard CTC, due to the uncertainty introduced by the wild-card symbol.

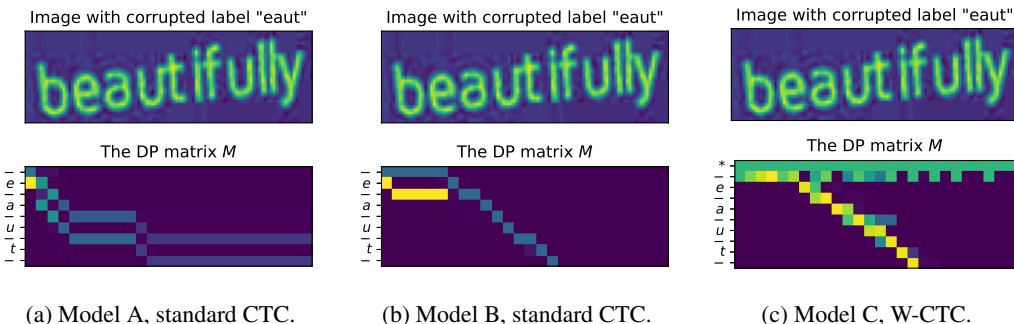

(a) Model A, standard CTC.    (b) Model B, standard CTC.    (c) Model C, W-CTC.

Figure 4: The alignment paths (in the trellis $M$) from CTC computation. Model A and C are trained on corrupted labels $r = 0.7$; model B is trained on clean data $r = 0$. Model A and B use standard CTC; model C uses W-CTC. Model A fails to produce correct alignment. Model B has correct alignment but with confusions. Model C provides clear path though trained on corrupted labels.

**Alignment Path:** To better illustrate how the partial alignment is done using the W-CTC loss, we provide a sample image with the corrupted label, and demonstrate how the DP trellis $M$ are different between using standard CTC and W-CTC. Recall that in Equation 3, $M$ provides a soft alignment path that generates the likelihood of $P(Y|X)$.

The sample image contains 11 characters, "beautifully". However, 70% of the label is masked and only 4 characters "eaut" are left to form the corrupted label. We feed the image and the corrupted label into three models: (1) Model A, trained using standard CTC, on the training data with labels 70% masked ($r = 0.7$). According to Figure 3, when $r = 0.7$, model A would learn nearly nothing from the data. (2) Model B, trained using standard CTC, but on the clean training data ($r = 0$). This model successfully learn the representations and should be able to predict the characters correctly. But it will face the difficulty dealing with the partial label here. (3) Model C, trained using W-CTC, on the training data with label 70% masked ($r = 0.7$). Figure 3 shows that the model using W-CTC

can still learn from the heavily corrupted ($r = 0.7$) labels. In Figure 4, we show the trellis $M$ for all three models. Here model A and B are using standard CTC's trellis, while model C continues to use the W-CTC's. So the augmented label of model A and B, is "-e-a-u-t-" ("-" is blank), while model C has the label prepended by a wild-card, i.e. "∗-e-a-u-t-".

In Figure 4, the lighter the color, the higher probability values. The nodes with higher values, form a soft alignment path that maps the input image to the corresponding label. As expected, model A (in Figure 4a) does not make reasonable predictions, therefore the path does not reflect the alignments between the partial label and the input image. In contrast, since model B is trained using clean data, it is able to get a reasonable path aligning 'eaut' to the image (shown in Figure 4b). Model B also maps the front of the image (containing letter "b", which is not in the corrupted label), to the "blank" symbol (the 1st and 3rd rows). However, it clearly struggles in the beginning (the lighter nodes in the left), and does not have a strong confidence (the path's color is darker). The final model C, although trained on the corrupted data, still generates a very reasonable alignment path in Figure 4c. The partial label is well aligned with the middle of the image, where the first row corresponds to the wild-card symbol that has a constant probability 1. Unlike model B, there are no confusions of other paths in Figure 4c. Considering that model C is trained purely on heavily corrupted data and only see 30% of training labels, the proposed W-CTC is very effective.

### 3.3 CONTINUOUS SIGN LANGUAGE RECOGNITION (CSLR)

Sign Language is a visual language composed of hand shape, hand movements, facial expression, mouth, head movements, etc. It is the most important communication medium for people having listening disabilities. In computer vision domain, researchers focus on recognizing a sequence of sign glosses from continuous sign language videos. Sign glosses refer to the spoken language words that represent the signs' meanings. This is called the Continuous Sign Language Recognition (CSLR) task, which has attracted lots of efforts, e.g., (Koller et al., 2019; Niu & Mak, 2020). Typically, CSLR models use CTC loss because no temporal alignment is provided in the data.

In this experiment, we leverage the framework proposed by Camgoz et al. (2020), and perform the CSLR (sign to gloss) task, with standard CTC and the W-CTC loss during training. The model is the encoder of Sign Language Recognition Transformer (SLRT) (Afouras et al., 2018). The dataset is PHOENIX14T (Camgoz et al., 2018), which provides both spoken language translations and gloss level annotations for weather broadcasts. Word-error-rate (WER) is used to measure the recognition quality on the test set. The results are presented in Table 1.

Table 1: WER on CSLR task. W-CTC significantly outperforms CTC when label corrupted.

| mask ratio $r$ | 0.0 | 0.1 | 0.2 | 0.3 | 0.4 | 0.5 | 0.6 | 0.7 | 0.8 | 0.9 |
|---|---|---|---|---|---|---|---|---|---|---|
| CTC | **0.286** | 0.422 | 0.537 | 0.663 | 0.735 | 0.814 | 0.896 | 0.945 | 0.989 | 0.981 |
| W-CTC | 0.297 | **0.328** | **0.346** | **0.395** | **0.418** | **0.392** | **0.440** | **0.568** | **0.926** | **0.916** |

From Table 1, we can see similar results as before. Only when data is absolutely clean ($r = 0$), standard CTC is slightly better (0.286 vs 0.297 WER). Starting from $r = 0.1$, the WER of the standard CTC model, deteriors quickly. On the contrary, W-CTC model can still achieve good WER up to $r = 0.6$. That means, even only trained on heavily corrupted labels, the model can still learn the representation of videos very well using the W-CTC loss.

### 3.4 ABLATION STUDY ON CHOICE OF $\sigma\{\mathcal{L}_{\text{CTC}}^{(j)}\}$

In this section, we conduct ablation study on how to combine the last two rows in the trellis of W-CTC computation. In other words, the choice of $\sigma\{\mathcal{L}_{\text{CTC}}^{(j)}\}$. We use the weighted sum, illustrated in Equation 4, in previous experiments. The other options could be taking the maximum or summing the probabilities. We formally define these three choices as:

$$\sigma\{\mathcal{L}_{CTC}^{(j)}\} = \begin{cases} \text{sum-prob (log-sum-exp): } -\log\left(\sum_j e^{(-\mathcal{L}_{\text{CTC}}^{(j)})}\right) = -\log\left(\sum_j P(Y|X[0:j])\right) \\ \text{max-prob (equivalent to } \min_j \mathcal{L}_{\text{CTC}}^{(j)}): \min_j \mathcal{L}_{\text{CTC}}^{(j)} = -\log\left(\max_j P(Y|X[0:j])\right) \\ \text{weighted-sum (Equation 4): } \sum_j w_j \mathcal{L}_{\text{CTC}}^{(j)}, \text{ where } w_j = \text{softmax}\{-\mathcal{L}_{\text{CTC}}^{(j)}\}[j] \end{cases}$$

(5)

where $N - 1 \leq j \leq T - 1$. Note that the 'sum-prob' means summing (over $j$) the probabilities of alignments between subsequence $X[0 : j]$ and label $Y$, i.e. $P(Y|X[0 : j])$. And 'max-prob' means selecting the $j$-th final node with the maximum probability among those alignments. So 'max-prob' is also equivalent to taking the minimum of $\mathcal{L}_{\text{CTC}}^{(j)}$. We perform 3 runs for each choice of $\sigma\{\cdot\}$, and present the average results on ASR, PR and CSLR tasks.

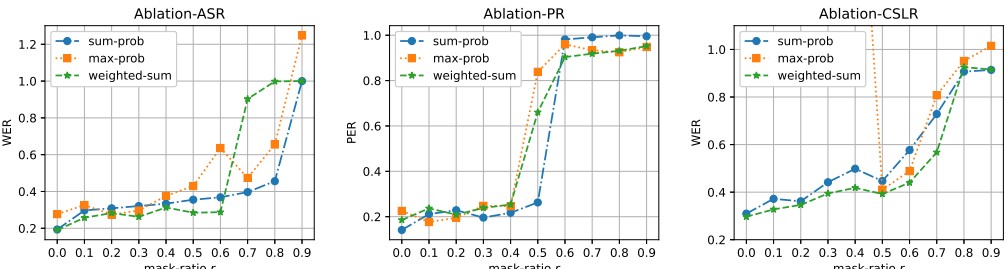

Figure 5: Ablation study on choices of $\sigma\{\cdot\}$. The 'max-prob' yields the worst performance in all three experiments. The 'weighted-sum' is slightly better than 'sum-prob' in general.

From Figure 5, we can tell that taking the max (orange curves) always yields worst performance (higher than other curves). Especially in the CSLR task, when mask-ratio $r < 0.5$, 'max' leads to a word-error-rate much higher than 1. The 'sum-prob' and 'weighted-sum' are similar but 'weighted-sum' wins by a small margin. For example, in ASR task, 'weighted-sum' performs better when $r < 0.7$; In PR task, all three curves are very close but 'sum-prob' is better only when $r = 0.5$; In CSLR task, 'weightes-sum' generally performs better across different $r$. Therefore, choosing 'weighted-sum' for $\sigma\{\cdot\}$ provides gains over other options.

## 4 RELATED WORK

Connectionist Temporal Classification (CTC) (Graves et al., 2006) was proposed as a loss function to train end-to-end sequence learning models. It is widely used in speech recognition, e.g., (Amodei et al., 2016; Battenberg et al., 2017); scene text recognition, e.g., (Graves et al., 2008; Shi et al., 2016); video action labeling, e.g. (Huang et al., 2016; Lin et al., 2017; Rouast & Adam, 2020); sign language translation, e.g., (Camgoz et al., 2020; Cui et al., 2017); and many other tasks.

To improve CTC, a variety of modified versions of CTC are proposed from different aspects. Auto Segmentation Criterion (ASG) is proposed in (Collobert et al., 2016) to simplify the CTC and remove the conditional independence assumption in CTC. In (Higuchi et al., 2020), the authors borrowed the idea from masked language model, and propose the Mask-CTC for non-autoregressive end-to-end ASR training. The work in (Lee & Watanabe, 2021) used CTC for intermediate layers as a regularization to the CTC training. Talnikar et al. (2021) proposed a joint training framework by taking advantages of both the unsupervised contrastive loss and the supervised CTC loss, for ASR. Entropy regularized CTC (Liu et al., 2018) was proposed to mitigate a problem that CTC often produces a peaky output distribution (Miao et al., 2015). A more recent work, Zeyer et al. (2021), provides a deeper numerical analysis on the role of the "blank" label and its resulting peaky behavior. Although CTC has not been applied to the partial-label training problem, a few research (Sakurai et al., 2007; Cai et al., 2019) in the time series domain, have studied a similar partial alignment problem. This paper is inspired by these works.

## 5 CONCLUSIONS

This paper studies the partial alignment (streaming) problem in sequence learning, where the label only matches a middle portion of the input sequence. To address this missing $Y$ problem, we propose the W-CTC loss, which improves the standard CTC loss with wild-cards. The wild-cards are prepended and also appended (implicitly done by considering the entire last two rows in trellis), to the original labels. The W-CTC is efficiently computed using the proposed scheme. Comprehensive experiments on a variety of speech and vision tasks, demonstrate that the W-CTC significantly outperforms standard CTC loss, when label has mismatches. Ablation study is conducted to verify our approach. Future work can implement more efficient backpropagation to speed up the computation.

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

## A  APPENDIX: EXPERIMENTS DETAILS

### A.1  CODEBASE AND HYPER PARAMETERS

Here we list the literatures and their code repositories that our implementation is based on, in Table 2. We also list the key training hyper-parameters in Table 3.

Table 2: List of code bases and their references that we use in the experiments.

| | | |
|---|---|---|
| W-CTC implementation | NA | https://github.com/vadimkantorov/ctc |
| ASR and PR experiments | Baevski et al. (2020) | https://github.com/huggingface/transformers |
| OCR experiment | Sun et al. (2020) | https://github.com/Media-Smart/vedastr |
| CSLR experiment | Camgoz et al. (2020) | https://github.com/neccam/slt |

Table 3: List of key training hyper-parameters.

| | Batch-size | Optimizer | LR | Steps |
|---|---|---|---|---|
| ASR | 32 | AdamW | 1e-4 | 7k (50 epochs) |
| PR | 32 | AdamW | 1e-4 | 7k (50 epochs) |
| OCR | 500 | AdaDelta | 1 | 150k |
| CSLR | 32 | Adam | 1e-3 | Stop if no better for 800 steps |

### A.2  SEQUENCE LENGTHS (DURATION) AND THEIR EFFECTS

In this section, we collected the statistics of the duration for ASR and PR experiments (the TIMIT dataset), as well as for CSLR experiment (the PHONEX14T dataset). We draw the histogram of the lengths (the number of frames, denoted as $T$) in Figure 6. The mean, std, max and min values are provided in Table 4. For OCR, we fix the image size as 30x100, so the duration (proportional to the image length, which is 100) is fixed.

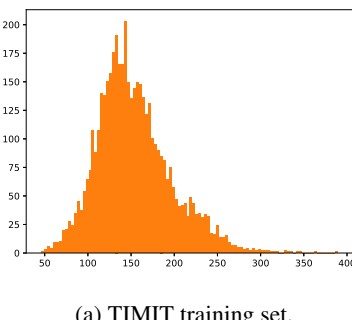
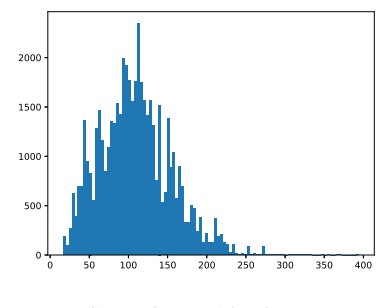

(a) TIMIT training set.    (b) PHONEX14T dataset.

Figure 6: The histogram of duration for TIMIT (for ASR and PR), and PHONEX14T (for CSLR) dataset.

Table 4: Duration (# of frames) for TIMIT training set (in ASR and PR experiments), and the PHONEX14T dataset (in CSLR experiment).

| | mean | std | min | max |
|---|---|---|---|---|
| TIMIT | 153.60 | 43.05 | 46 | 389 |
| PHONEX14T | 108.67 | 45.25 | 16 | 395 |

From Figure 6 and Table 4, we can see the $T$ varies a lot (e.g. 16 to 395). This covers a wide range of the duration. In theory, if $T$ is not normalized out, it should hurt the performance when we

use the proposed W-CTC without normalization. The longer sequence get higher probability (total probability is $2^T$), resulting in a smaller loss.

However, empirically we found normalization has little impact as shown in Appendix B, in presence of such wide range of duration in the dataset. The W-CTC performs constantly well. Note that in a seq2seq generative model, without normalization, the model tends to generate short sentences, which is unwanted. But here we are dealing with a different problem. The higher probability essentially reduces some weight to longer sequences. The model is encouraged to pay more attention to the correctness of the shorter sequences. However, this does not necessarily lead to poorer performance. Shorter audios could be either clear (easy to learn) or noisy (hard to learn), the same for longer sequences. Therefore, it is like bootstraping, but in a random fashion. It does not bring gains, neither hurts. Empirically we also didn't observe difference whether normalize or not. So we choose the unnormalized version due to simplicity. This is not theoretically proved, and requires tests on more datasets in future work.

### A.3 THE MASKING PROCEDURE

Let us use an example to illustrate the masking procedure. Suppose we have a label string "helloworld", 10 characters in total. We want to mask 50%, which means 5 characters. We only mask front and end, randomly. So we generate a random starting position, uniformly from $[0, 5]$. Suppose the random generator yields a value 3, then we keep the characters from 3 to 7, and throw the rest. In this case, the remaining label for training will be "llowo".

Masking only applies in training, as the goal is to let the model learn from corrupted labels. It is hard to find a public dataset that contains incomplete labels (they are mostly very clean), so we apply masking in the training phase to mimic the corrupted label scenario. The inference is just to test whether the model has successfully learned the representations, so there is no masking involved at inference time.

### A.4 EXPERIMENTS ON RANDOM MASK-RATIO

We conduct the ASR experiment on TIMIT dataset, using a random mask-ratio. This means, we do not fix the mask-ratio $r$, but for each utterance, we generate a random $r$ that is uniformly distributed between 0 and 1. Apparently, $\mathbb{E}[r] = 0.5$. We compare with the case that has a fixed mask-ratio $r = 0.5$, as well as no-masking ($r = 0$). Word-error-rate (WER) is obtained from 3 runs for each case. We also include standard CTC. The results (mean and std) are listed in Table 5.

Table 5: ASR experiments on random mask-ratio

| WER | W-CTC | | | CTC | |
|---|---|---|---|---|---|
| | random $r$ | $r = 0.5$ | $r = 0$ | $r = 0.5$ | $r = 0$ |
| mean | 0.260 | 0.285 | 0.191 | 0.789 | 0.189 |
| std | 0.013 | 0.005 | 0.007 | 0.183 | 0.005 |

As expected, the random $r$ has similar results as $r = 0.5$, because the expectation is $0.5$. Both are not far from $r = 0$ (clean label) case, which verifies the effectiveness of W-CTC. W-CTC are much better than standard CTC when $r > 0$.

## B APPENDIX: NORMALIZATION

### B.1 NORMALIZATION EFFECTS

Note that since an additional wild-card symbol "$*$", with a constant probability 1, is added in each frame, the total probability accumulating $T$ frames is no longer normalized. Specifically, $Z = 2^T$ is the normalization factor. Therefore, to make $P(Y|X)$ a proper likelihood, it should be normalized by $Z$, i.e. $P(Y|X)/Z$. Otherwise, the longer input sequence (larger $T$), the higher $P(Y|X)$ values it can get. Due to the fact $\mathcal{L}_{\text{CTC}} = -\log P(Y|X)$, the loss on longer sequence would be smaller than the shorter ones. In other words, it punishes more on shorter $X$, regardless of the prediction

quality. Clearly, this is not what we want. A mitigation to this, is to normalize the probability and obtain the normalized W-CTC loss:

$$\mathcal{L}_{\text{norm}} = -\log\left(\frac{P(Y|X)}{2^T}\right) \approx \mathcal{L}_{\text{W-CTC}} + 0.3T \tag{6}$$

From Equation 6, we can see that the longer sequence (larger $T$), the more punishment ( $\log_e(2^T) \approx 0.3T$ ) is added to the original W-CTC loss. However, we didn't observe a clear advantage to perform normalization in this way. In the following Figure 7, we compare the performance of unnormalized and normalized version of W-CTC. This is obtained from the average of 3 runs. It is very hard to observe a clear trend that the normalized version outperforms the unnormalized version. We have a more detailed-discussion in Appendix A.2. So in the experiments, we just use the unnormalized W-CTC for simplicity.

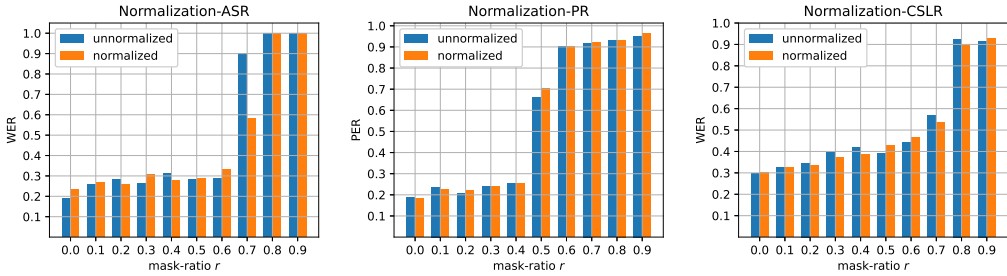

Figure 7: Compare unnormalized loss with normalized version. No significant difference is observed.

## B.2 THE EMISSION PROBABILITY OF WILD-CARD

In this paper, we set the probability of the wild-card to be always 1, meaning that it matches any character in the labels. However, making $P(*|X) = 1$, and $\sum_s P(s|X) = 1$ (where $s$ represents any possible character in vocabulary) at the same time, leads to a total probability 2 instead of 1 (unnormalized), for each frame. One way to mitigate this issue, is to normalize over $T$ (the total number of frames), which is studied in Appendix B. Here we provide another solution, that making $P(*|X) = p$, and all the characters get a total probability $\sum_s P(s|X) = 1 - p$. The obvious advantage is that we get a normalized probability for each frame, thus the maximum likelihood estimation could be carried out without concerns. On the other hand, the disadvantage is that we have to select $p$ as a hyper-parameter, which requires tuning for each experiment. Meanwhile, it slightly alters the meaning of the wild-card. For each frame, we assign a probability less than 1, to the wild-card. This makes the wild-card being treated like a regular symbol in the vocabulary.

Nevertheless, we conducted an additional experiment to study whether this scheme works in practice. We range the value $p$ from 0.1 to 0.9. For each $p$, we train the model under different mask ratio $r \in [0, 1)$. We perform the ASR experiment here. All the other hyper-parameters remain the same.

Figure 8 show the results on ASR experiment. The blue solid curve (labeled as "proposed") is the proposed W-CTC, with constant probability 1 for the wild-card (the unnormalized version). All the other dotted lines represent different hyper-parameter $p$, from 0.1 to 0.9. We plot the mean and std of word-error-rate (WER), against the mask-ratio $r$. The lower, the better. It appears that $P(*|X) = 0.8$ is slightly better compared with other cases, only when $r > 0.6$. However, the difference is subtle, and all curves are very similar. The unnormalized version (proposed) performs as good as others.

Considering the hyper-parameter needs to be tuned, and the meaning of wild-card is slightly changed, there is no clear advantage using this scheme to control the wild-card's emission probability, at least from an empirical point of view. Therefore, combined with this study and the one in Appendix B, we choose the unnormalized version due to its simplicity. However, for a better understanding, more tests on a wider range of dataset needs to be carried out. It is worth noticing

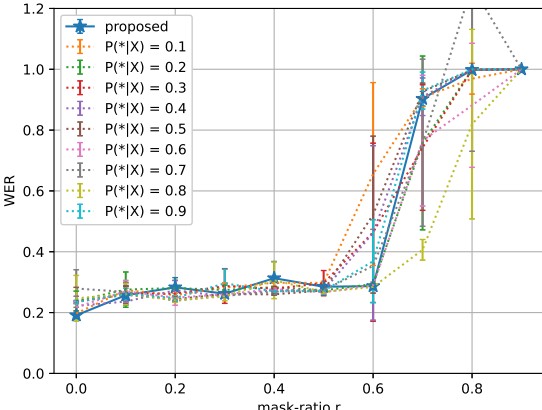

Figure 8: The ASR experiment, when $P(*|X) = p$ and $\sum_s P(s|X) = 1 - p$. The plots are word-error-rate vs mask-ratio.

that, this scheme could also brings the potential of dynamically adjusting $p$. Instead of a fixed $p$, it could be learned by the network and adjusted for each individual frames. This will again bring more complexity but can potentially boost the performance, and avoid the hyper-parameter tuning.

### B.3 DISCUSSION ON WILD-CARD AND WEIGHTED-SUM

The proposed W-CTC, essentially enables wild-card at both sides. However, they are done in two different ways. For the beginning, we explicitly prepend the wild-card symbol (*) before the actual label. For the ending, we let the trellis ends at an arbitrary node in the last row. This is adopted from the SPRING algorithm proposed in (Sakurai et al., 2007), originally for DTW. We would like to explain the rationale for the two different choices.

**Why wild-card only in the beginning:** Prepending a wild-card in front, allow the path to start in the middle of the frame. However, appending a wild-card in the back, is slightly different. Recall that each node in the trellis represents the probability of aggregated paths arriving at that node. If the the trellis' last row corresponds to the wild-card, whenever a node in the last row is reached, it will carry its probability to the next node (its neighbor in the right) without penalty. In the final node (the right-most node in the last row), it essentially sums all the possible ending-positions' probability. This is conceptually equivalent to the "sum-prob" case in Equation (5). However, we may want to choose the best possible ending point, maximize its probability. Therefore, using the wild-card loses the flexibility of hard-max or soft-max, as opposed to the weighted-sum scheme. Another issue comes with the unnormalized probability. If the ending wild-card node propogate to the next node (the neighbor in the right), the total probability increases by 1 for each step towards the right-most node. The paths probability is typically a small number, because there are a huge number of paths. In this unnormalized version, the paths' probability is overridden by the wild-cards' probability, resulting in a failure of training. We conducted experiments and observed that when having wide-cards at both sides, the training does not converge. To mitigate this, using a normalized version could be a solution, but brings more complexity as described in the paper. It is much simpler to use a weight-sum scheme in the end rather than the wild-card.

**Why weighted-sum:** Ideally we would like to choose the ending node with the maximum aggregated path probability. However, empirically we found that taking the hard maximum, leading to convergence issues. Therefore, a relaxed version, the soft-max (the weighted-sum) is adopted here. Our ablation study in Section 3.4 also verifies this choice. Intuitively speaking, the hard choice (take the max one), will ignore the frames beyond that ending point. For example, if we have 10 frames, and the path ending at the 8th frame, got the maximum probability. Once we use max-prob, we will completely ignore the 9th and 10th frame, where the gradient will not back-propogate to these 2 frames. The soft version (weighted-sum) mitigate this problem to some extent. A theoretical justification is due to future work.

**Why weighted-sum only at the end:** The weighted-sum scheme is not straight-forward to be applied in front. This is to select the best path. There is no path in the beginning phase. If we want to choose (a hard selection) a starting position, it still needs to wait until the very end to get a metric, e.g. the path's probability, in order to make the selection. Meanwhile, hard selection also brings convergence issues. A weighted combination of starting point has the same issue that it has to wait until the path ends. In contrast, a wild-card is both conceptually simpler and easier for implementation, to handle the front part.

## C  APPENDIX: CONVERGENCE BEHAVIOR

### C.1  TRAINING CURVES COMPARISON

Since W-CTC expand the number of possible alignment paths, it is worth investigating the convergence behavior of W-CTC, compared with the standard CTC. In this subsection, we draw the training curves (evaluation word-error-rate vs training steps) to show that, the proposed W-CTC converges as good as standard CTC, but achieves much better results when label is incomplete. The ASR and PR experiments are performed. The batch size and the learning rate are the same for both W-CTC and CTC. Both are running 7000 steps (50 epochs).

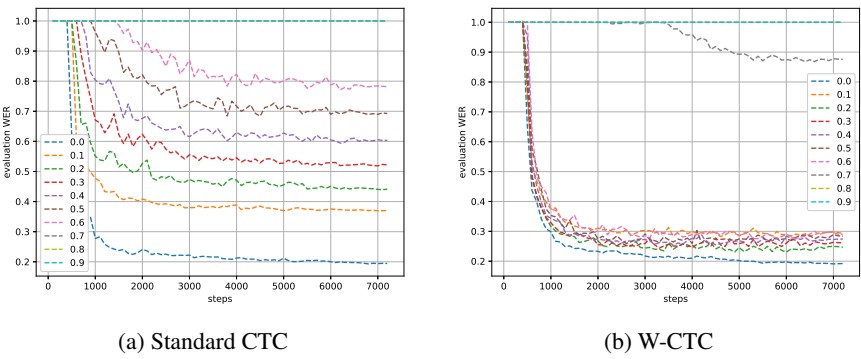

(a) Standard CTC                              (b) W-CTC

Figure 9: ASR experiments. WER vs training steps.

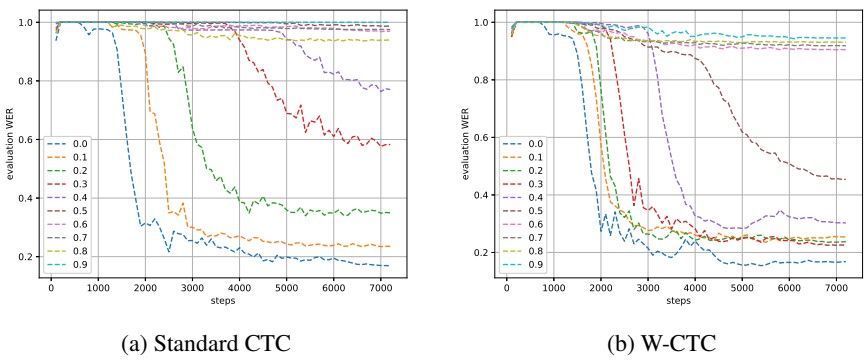

(a) Standard CTC                              (b) W-CTC

Figure 10: PR experiments. WER vs training steps.

Figure 9 shows the convergence behavior for the ASR experiment, while Figure 10 for the PR experiment. Each curve represents a training dynamics for a particular mask-ratio $r$. Clearly, in both experiments, whenever label is not full ($r > 0$), W-CTC not only yields a better final WER, but also converges faster than standard CTC. For example, when $r = 0.6$, in Figure 9 (a), the WER starts decreasing after around 1500 steps (the pink curve). However, in Figure 9 (b), it starts decreasing at around 500 steps, which is nearly 1000 steps earlier than the standard CTC. When label is clean (the blue curve), both standard CTC and W-CTC converges almost identically. The PR task is harder, but the same trends hold when comparing Figure 10 (a) and (b). Therefore, we can conclude that,

W-CTC converges as good as standard CTC, when label is clean, and becomes much better when label is incomplete.

## C.2 TRAINING LOSS COMPARISON

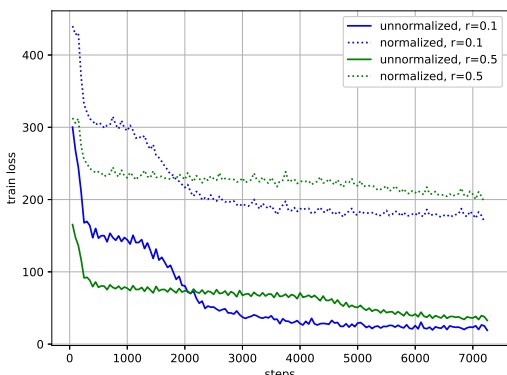

Figure 11: The training loss comparison between the unnormalized version and the normalized version.

Since Appendix B and Appendix A.2 suggests, the normalized W-CTC loss will be the original proposed W-CTC loss, plus a $0.3T$ term, where $T$ is the number of input frames. To make a comparison with the model trained using the normalized version ($+0.3T$), with the unnormalized version, besides the performance comparison done in Appendix B, we also plot the training loss here to verify if the additional term $0.3T$ will have significant impact over the training. We pick the case of mask-ratio $r = 0.1$ and $0.5$, on the PR task (using TIMIT dataset).

In Figure 11, the solid lines (the two curves at the bottom) corresponds to the unnormalized version, where the dashed lines (the upper two curves) are normalized ($+0.3T$). The blue curves are the case of $r = 0.1$, where the green curves are $r = 0.5$. We can see that in both cases, the curves between nromalized and unnormalized, are very similar except for a vertical shift. The unnormalized version are higher due to the $+0.3T$ term. Therefore, it shows that the normalization term here does not affect the training trend in general, but simply adding a constant values on top.

## D APPENDIX: MORE PHONEME RECOGNITION EXPERIMENTS

### D.1 PR ON REDUCED PHONEME SET

It is common in literature, e.g., (Lopes & Perdigao, 2011), (Lee & Hon, 1989), (Baevski et al., 2020), to use the reduced phoneme set to test the Phoneme Recognition performance. The reduced set maps the original 61 phonemes into 39 classes. The mapping table in (Lopes & Perdigao, 2011) is provided below. The original phoneme "q" is removed in the reduced set.

Table 6: Mapping table from 61 phonemes to 39 phonemes to form the reduced set.

| aa,ao | ah,ax,ax-h | er,axr | hh,hv | ih,ix | l,el | m,em | n,en,nx | ng,eng | sh,zh | uw,ux | pcl,tcl,kcl,bcl,dcl,gcl,h#,pau,epi |
|-------|------------|--------|-------|-------|------|------|---------|--------|-------|-------|-----------------------------------|
| aa    | ah         | er     | hh    | ih    | l    | m    | n       | ng     | sh    | uw    | sil                               |

We train the model using original 61 phonemes, and evaluate on the reduced 39 phonemes set, by following the mapping in Table 6. The average performance over 3 runs, is reported in Figure 12 below. The results are similar as the experiments on the full 61 phone set, which is shown in Figure 2's right panel. With standard CTC loss, the trained model's performance drops quickly as the mask-ratio $r$ increases. In contrast, the W-CTC helps the model learn from corrupted labels and yield consistent performance until $r > 0.6$.

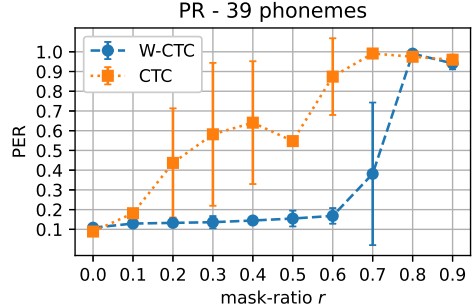

Figure 12: PER vs mask-ratio, on reduced phoneme set (39 phonemes) of TIMIT test set.

## E PR EXPERIMENT USING CRDNN BACKBONE

In the ASR and PR experiments, we adopt the pretrained Wav2vec-2.0 model because it is one of the state-of-the-art model that yields best ASR and PR results on TIMTI dataset. One concern is that, since the pretrained Wav2vec-2.0 has learned the representations during its unsupervised pretraining, the fine-tuning on TIMIT dataset may just learn a simple mapping rather than a representation of the acoustic signals. To address this concern, we conduct another experiment using a customized CRDNN (CNN + RNN + MLP) model, trained from scratch with the W-CTC loss, to demonstrate that our proposed W-CTC is model agnostic and effective in the learning of representations.

The model and codebase are from SpeechBrain [1]. We use the same hyper-parameters as shown in the repository's config file, except setting the learning rate to 0.1, the patience to 10, and train 100 epochs. The reason is that making the learning rate to 1, sometimes cause unstable training and yields NaN loss. To speed up the training with lr=0.1, in the scheduler, we increase patience to 10 such that the learning rate can stay constant for more epochs.

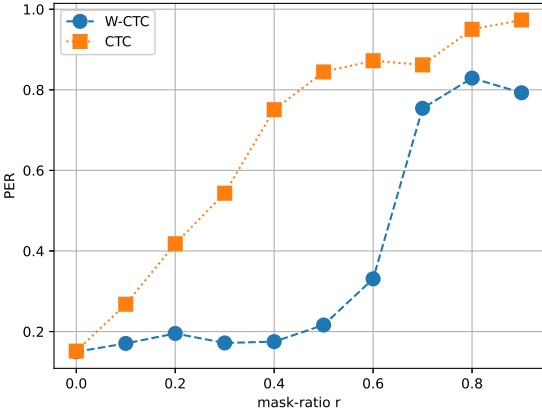

Figure 13: The PR experiment on 39 phonemes in TIMIT dataset, using the CRDNN model.

Figure 13 shows the results obtained from standard CTC and proposed W-CTC training, under different mask-ratio $r$. Note that the reported standard CTC training results on the website (when $r = 0$) is 0.148, where we got 0.150 (the left most dot on orange curve). The curves are very similar to the Wav2vec-2.0's. As $r$ increases, the standard CTC loss's performance drops almost linearly. In contrast, the W-CTC can stay constant up to $r = 0.6$. Overall, the W-CTC outperforms the standard CTC by a large margin. This is consistent with the Wav2vec-2.0's results, as well as the other two

---

[1] https://github.com/speechbrain/speechbrain/tree/develop/recipes/TIMIT/ASR/CTC

experiments (the OCR and CSLR models are also trained from scratch). Therefore, we conclude that W-CTC is able to enable representation learning, rather than a simple mapping.

## F    APPENDIX: MORE OCR EXAMPLES

This subsection provides more examples to illustrate the difference in the trellis, among W-CTC and the standard CTC. Following Section 3.2, we use the images from the OCR example. We randomly select a few images with long labels (length ¿ 10) for a better illustration. We still have three models. The first one is the standard CTC, trained on corrupted labels where mask-ratio $r = 0.7$. We plot them in the left column. The second model is the standard CTC, but trained on clean labels. They are expected to learn the representations, but will face the difficulty in the corrupted label cases. We plot them in the middle column. The last model is the W-CTC, trained on corrupted labels $r = 0.7$. We plot them in the right column.

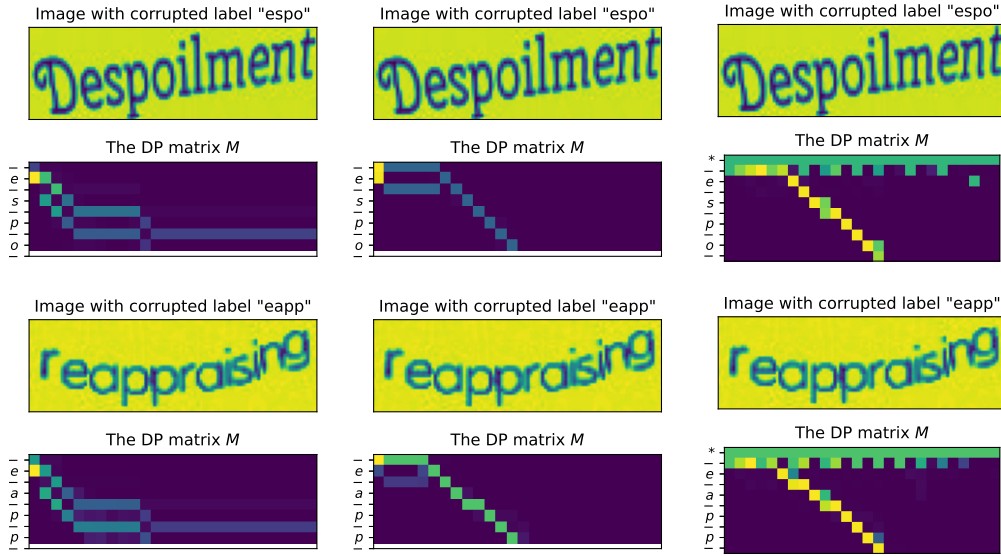

Figure 14: Left: Standard CTC trained on mask-ratio $r = 0.7$; Middle: Standard CTC trained on clean label; Right: W-CTC trained on $r = 0.7$. Here the standard CTC only make reasonable inference if it is trained on the clean label, where W-CTC generally output correct paths.

We group the images into 3 categories to show different aspects of the standard CTC vs W-CTC:

In Figure 14, we show two examples that have very clean images. We have the same results as shown in Section 3.2. The standard CTC trained on incomplete labels, has no clue how the alignment is done in these examples. The standard CTC trained on clean labels, on the contrary, can figure out the correct path, although with very low confidence. The W-CTC can obtain the strong paths that correctly reflect the alignments, even though it is trained on heavy corrupted labels.

In Figure 15, the images becomes more complicated (has curves or less clear). The missing part in the label is pushed back towards the beginning of the labels. Now, even trained on the clean label, the standard CTC is no longer able to identify the correct paths (the middle column). However, the W-CTC is hardly affected, and still generate the main path that corresponds to the alignments. There are some tiny confusion paths, due to repeated characters in the labels.

In Figure 16, we show the images with high confusions. The remaining labels has overlap to more than one places in the figure. For example, the "st" appears twice in "mistrusting". These lead to a second path with high confidence, for the W-CTC (the right column). However, before those confusion paths reach their ends, they got cut off due to mismatches. Therefore, the W-CTC can still figure out the most likely path that often corresponds to the true alignment. The standard CTC does not generate any reasonable paths in these examples, no matter it's trained on incomplete labels (left column), or clean labels (middle column).

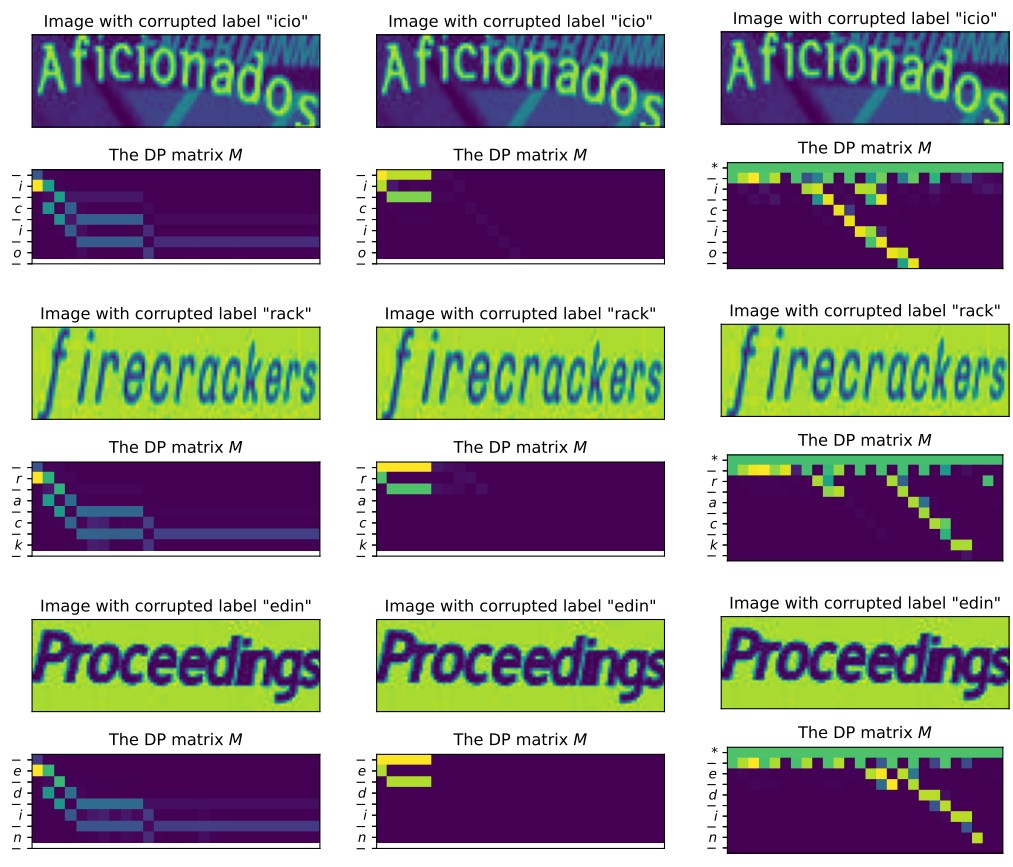

Figure 15: Left, middle and right are the same as above. Here the standard CTC trained on clean label, cannot even make correct paths when facing such corrupted label. W-CTC is not affected.

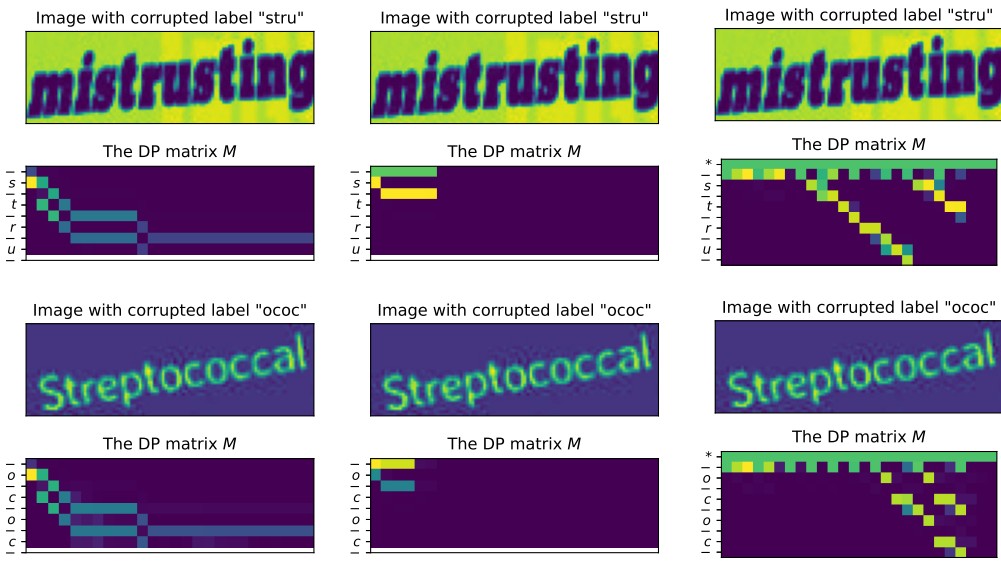

Figure 16: Left, middle and right are the same as above. These are confusing examples. For example, the "st" appears twice in "mistrusting", the "oc" appears twice in "streptococcal". We can see clean paths starting from the incorrect starting positions, in the W-CTC figures (right plots). But those incorrect paths end before reaching the end. Standard CTC cannot handle them at all.

In conclusion, we can see a clear advantage of the proposed W-CTC, when dealing with incomplete labels. Due to the wild-cards, it can match part of the input sequences (here are the images), and generate reasonable alignment paths. The standard CTC, even trained on clean labels, can only output reasonable path, if the remained labels corresponds to the beginning of the sequence. If the front part is missing, the standard CTC completely fails. These examples verify the effectiveness of the proposed W-CTC method.

