# OpenReview forum: "W-CTC: a Connectionist Temporal Classification Loss with Wild Cards"
_ICLR.cc/2022/Conference — ICLR 2022 Poster_

### Official Review · Reviewer_K22X · 2021-11-01

**Correctness:** 4
**Technical Novelty And Significance:** 2
**Empirical Novelty And Significance:** 2
**Recommendation:** 6
**Confidence:** 5

**Main Review:**

Strengths:

1. This paper describes the algorithm in a very clean way. It is very easy to understand the algorithm.

2. The proposed method is evaluated on three different domains. This helps to support the claim of the paper better.

Weakness:

1. The experiments are conducted only with faked masked data. Although it is valuable to have a paper discussing this topic, it is not very clear the actual benefit in real scenarios. Given that the technique itself it not very novel, this might be important to claim the novelty.

2. It uses a wild-card for the beginning while it uses unconstrained endpoints for the ending. There is no explanation about the choice. It would be great why the combination is chosen.

3. The proposed method is slightly worse than the standard CTC at r=0.0. That means that it would not be a drop-in replacement of the standard CTC. It would be better if it was shown that the proposed algorithm was actually useful for some real application.

**Summary Of The Paper:**

This paper proposes a method to utilize partially labeled data for the CTC loss. In order to handle untrascribed lables, it introduces a wild-card for the beginning of the sequence and uses unconstrained endpoints for the ending of the sequence. It also discusses how to summarize the paths at the ending. The proposed algorithm was evaluated with simulated masked data and it was shown that the proposed algorithm can handle incomplete labels more effectively than the naive CTC loss while the naive CTC loss is slightly better when the label is complete.

Dynamic time warping (or DP matching) with unconstrained endpoints itself is not a new idea and a classical topic for speech recognition (e.g. word spotting). The contribution of the paper is the formal introduction of the approach to CTC and to give experimental results to confirm the effectiveness.

**Summary Of The Review:**

The paper is clean and it is easy to understand the content. The experiments are fair and the effectiveness of the approach is confirmed using simulated data. It is great that the results are given for three different domains. There are certain weak points in the paper, but the strengths slightly outweigh the weaknesses. I would recommend a weak accept.

---

> ### Author Response · Authors · 2021-11-15
> **Thank you so much for your positive feedback!**
>
> Thank you so much for your positive feedback and we would like to answer your questions as follows:
>
> 1. The experimental data is fake masked.
>
> A: In fact, this paper is motivated when we tried to preprocess (segment) real data. But for paper writing, we have to run on public datasets, which are mostly clean. It is a bit difficult to find public dataset with incomplete labels for training, where testing labels are clean. The data itself are not synthetic but labels are indeed randomly masked. As pointed by reviewer sfbX as well, the method would help a lot when we lack of an accurate aligner, which often happens in practice.
>
> 2. Why wild-card in the begining but unconstrained endpoints for the ending ?
>
> A: Let us answer in two folds:
>
> (A) Why wild-card only in the beginning: Prepending a wild-card allows flexibility to start from a middle frame. However, appending a wild-card has two main issues. We provide a detailed explanation in Appendix D.6. In short: (1) It is equivalent to the 'sum-prob' case in Equation 5. So it loses the flexibility to use "max-prob" or "softmax-prob" (the weighted-sum), if we want to select the best ending position. (2) In our unnormalized version, this scheme will accumulate probabilities from wild-cards, and override the actual paths' probability. Just like gradient vanishing problem. We observed that it does not converge if wild-card at both sides using current settings. We can use the normalized version to mitigate, but it becomes more complex than just using weighted-sum at the end.
>
> (B) Why unconstraint point only in the end: It is not straight-forward to apply this in front, as it is to obtain the aggregated paths' probability. There is no path in the begining. If choosing a starting position based on some metric, e.g. path probability, the metric still needs to be computed at the end of the path. In contrast, wild-card is both conceptually simpler and easier for implementation, to handle the front part. We updated this discussion in Appendix D.6.
>
> 3. WCTC is slightly worse than standard CTC when r = 0.
>
> A: This is a very good point. We suggest one way to mitigate, is to adopt a hybrid fashion. When we are confident that the collected data is clean, we can use the standard CTC. But when we don't have an accurate alginment, we can simply leave some redundancy when doing segmentations, and run WCTC on those collected data that having gabbage infomation at both ends.
>
> Thanks again for the valueable comments. We put all additional experiments/discussions during the response period, in a big Appendix D in the paper, for easy to look up. In the final version we will merge that into different parts of the paper.

---

> > ### Comment · Reviewer_K22X · 2021-11-21
> > **Thank you for the response.**
> >
> > Thank you for the response.
> >
> > Now, I have a better understanding on the uses of the wild-card and the unconstrained endpoints at the beginning and ending, respectively.
> >
> > After thinking more about this problem, I find that the paper provides an empirical method to deal with the unnormalized score caused by the wild-card and the unconstrained endpoints. Probably, it cannot be simply addressed by normalizing the score by just computing Z for the graph as the summation to compute Z is not mathematically correct (i.e. sub-paths of a path in the set of all paths to compute Z are also members of the set). This is probably the reason why W-CTC is worse at r=0.
> >
> > I think it is possible to design an algorithm that does not have the normalization issue. This would be good future work.

---

> > > ### Author Response · Authors · 2021-11-21
> > > **Glad we answered your questions.**
> > >
> > > Dear Reviewer,
> > >
> > > We are glad that we addressed your questions.
> > >
> > > We also believe that, as discussed with the reviewer pcpu as well, the normalization part could be an interesting topic due to future study. We appreciate your constructive comments and feedbacks during this response period. Thank you!

---

### Official Review · Reviewer_pcpu · 2021-11-02

**Correctness:** 3
**Technical Novelty And Significance:** 2
**Empirical Novelty And Significance:** 3
**Recommendation:** 6
**Confidence:** 4

**Main Review:**

**Strong points**
- Authors propose simple and efficient modification in CTC algorithm to support partial labels (continuous)
- Method is proved empirically to be applicable across different domains/tasks (OCR, ASR, CSLR) for a large percentage of label corruption with significance testing.

**Weak points**
- mostly the idea is based on SPRING (dynamic time warping, DTW) which is mentioned in the paper and applied for CTC instead of DTW.
- Wav2vec is learning actually representation of phonemes/tokens and later fine-tuning is more about mapping each representation cluster to the proper label. That is why it could be that W-CTC could be enough to simply learn this mapping but not the necessary representations.
- (Limitation) Algorithm doesn't work with label corruption in the middle or at random places, which is the more often case during human labeling process.

The paper is very well written and presented: all necessary intro information is given, there are nice figures to present comparison between CTC and W-CTC are given. Overall, it is very simple to read and follow the whole text and statements. I have only minor comments:
- I suggest to mention also ASG loss https://arxiv.org/pdf/1609.03193.pdf as one of the directions on CTC modification
- typo: "The average performance over 3 runs, are reported" -> "The average performance over 3 runs, is reported"
- Could authors provide more details on how the masking is done: random val for the begging and (r - val) for the end?
- To resolve concern pointed above on wav2vec: Could authors perform additional experiments (even without statistical significance) for ASR where model is trained from scratch and not pretrained wav2vec to demonstrate that W-CTC is able to learn proper representations too (i believe it will as it is doing well in OCR and CSLR tasks)?
- Why in Eq. (7) it is $-0.3T$ and not $+0.3T$?
- Does normalization from Eq. (6) not converge with all ways in Eq. (5)?
- I don't get why in Eq. (6) there is $2^j$ and not $2^T$ as we assume that at the end we are staying in the wild-card state. Then actually Eq. (7) is the explicit form for normalized W-CTC.
- With respect to results on the normalization: what is the average and std for the T in different tasks? Could it be that variation is very small so that normalization doesn't influence?


**Summary Of The Paper:**

Mapping between two sequences of different length is frequently done via Connectionist Temporal Classification (CTC) loss when alignment is not available. CTC performs the full alignment between input and label sequences. However, there are applications when label is incomplete and only part of it is given. In the current paper authors consider the case when begin and end of label sequence are missed and extend CTC applicability to these partial labels. Based on the dynamic time warping SPRING algorithm for incomplete labels authors propose W-CTC: they similarly prepend and append label with "*" token which model missed begin and end of label sequence; then they aggregate all valid paths with the latter modification. W-CTC is empirically proved to significantly improve performance over CTC even if up to 40-70% label sequence is missed (overall performance similar to the complete label case) across different tasks, like speech, optical character, and continuous sign language recognition. W-CTC is simple and efficient.

**Summary Of The Review:**

The paper is very well written with clear explanation of proposed method, well covered experimental results across domains and tasks. Proposed idea is mainly based on existing DTW algorithm (SPRING), however it is extended and applied to CTC demonstrating significantly improved results on incomplete labels. Thus, I would recommend this paper to be accepted. It could lead to a broader discussion on methods with incomplete labels and data - more real case scenario of data collection.

---

> ### Author Response · Authors · 2021-11-16
> **Thank you for your positive feedback!**
>
> Thank you so much for your positive feedback and we would like to answer your questions as follows:
>
> 1. Wav2vec2 is too powerful such that wctc only learns a mapping rather than representations.
>
> A: This is a good point. Though on other two tasks (OCR and CSLR), the models are trained from scratch, in the ASR and PR experiment, we use the pretrained wav2vec2, simply because it is the SOTA model. But we are glad to conduct additional experiment to verify this, in Appendix D.7. Here we train a CRDNN (CNN+RNN+linear) model from scratch, to perform phoneme recognition on the same TIMIT dataset. The codebase we use is from SpeechBrain (https://github.com/speechbrain/speechbrain/tree/develop/recipes/TIMIT/ASR/CTC). We modify the CTC part and perform the same experiments as in the paper. The results show consistent performance as in wav2vec2, though generally worse due to the model's capacity. This, along with the other two experiments, can verify that WCTC can enable the model to learn representations, and performs better than standard CTC in missing label cases. Please refer to Appendix D.7 for details.
>
> 2. The limitation that only handles missing label at both sides.
>
> A: Yes this is one major limitation stated in the paper as well. But as pointed out by reviewer sfbX, we often face segmentation problems that no accurate alignment is available. With WCTC, we can simply have some redundancy and still make model training possible.
>
> 3. Mention ASG loss as one important CTC modification.
>
> A: Agreed. ASG (Auto Segmentation Criterion) is indeed one of the most popular modifications (simpler and more effective) of CTC. Thanks for the suggestion and we have updated the paper accordingly.
>
> 4. Typos.
>
> A: Thanks for letting us know. We have updated accordingly.
>
> 5. How the masking is done?
>
> A: We applogize for not making this clear in the paper. Yes we use a random val for begining. For example, suppose we have a label string "helloworld", 10 characters in total. We want to mask 50%, which means 5 characters. So we generate a random starting position, uniformly from [0, 5]. Suppose we get 3 from rand function, then we keep the characters from 3 to 7, and throw the rest. In this case, the remaining label for training will be "llowo". We updated this part in Appendix D.1.
>
> 6. Why Equation 7 is -0.3T but not +0.3T ?
>
> A: Sorry this is a typo, it should be +0.3T. Intuitively, the longer sequence, the larger total prob, which means the smaller loss, so we need to compensate for that by adding more (+0.3T) into the loss. We have corrected it. Thanks for pointing out.
>
> 7. Does Eq 6 not converge with all choices listed in Eq 5 ?
>
> A: We tested Eq 6 with all three schemes, unfortunately none of them converges (resulting WER > 1).
>
> 8. Why Eq (6) has 2^j rather than 2^T as we assume ending at wild-card state ?
>
> A: This is a very good point. The relaxed max scheme (weighted-sum) still makes a selection on where to end. If ending at j-th frame (j < T), the length shoudl be j and total prob is 2^j. But from another perspective, just as you pointed, if considering ending at j-th frame is in the wild-card state, which actually lasts until the final frame T, then the total prob should be 2^T. My guess was both make senses. But the experiments show that the first view does not work. Therefore, at least empirically, your point (the second view) is indeed a better understanding of the ending state. We have updated this part in Appendix C accordingly.
>
> 9. What is mean and std for T in different tasks. Are they too small to make difference on normalization ?
>
> A: This is a great point! We acknowledge we didn't pay attention to the range of durations. We collected the durations and draw the histogram in Appendix D.5. At least for the ASR, PR and CSLR experiments, the range of duration is not small (min is 16, max is 395, std is 45 frames). We suggest the effect of T is orthogonal to the performance. It encourages the model to pay more attention to short utterances. Short audios could be either noisy (hard to learn) or clean (easy). That could be the reason why we didn't observe difference whether normalizing T. But this is not theoretically proved. Empirically we favor the unnormalized version due to simplicity.
>
> We highly appreciate these insightful comments and suggestions. We have updated the paper based on them, and answered questions above. We put all additional experiments/discussions during the response period, in a big Appendix D in the paper, for easy to look up. In the final version we will merge that into different parts of the paper. We are happy to have further discussions.

---

> > ### Comment · Reviewer_pcpu · 2021-11-18
> > **Discussion on normalization**
> >
> > Dear authors,
> >
> > Thanks for all clarifications and additional experiments including training from scratch without wav2vec. All additional experiments and comments make sense to me and look strong and reasonable.
> >
> > Now, I have only one comment regarding normalization.
> >
> > I would say that normalization should be computed not with respect to one path and depends on its length, but you need to compute the entire global normalization of the entire graph (so sum up all paths together). In the latter case (which is correct one) I guess you will get different normalization constant, not the $2^j$. That is why simple view is to treat that we stay in the wild card state at the end having $P=1$ and then the global graph normalization is exactly $2^T$ and we do normalization to it. This could explain why with eq. (6) you observe divergence while with $2^T$ you get convergence. One thing I still have no idea, why without any normalization it still converges even with the duration variation in the data. Do you do any loss normalization by input duration (in some implementation people do CTC loss division by the input/target duration) per sample (this could explain why it still works)? Would be interesting to have a look at the loss magnitude over the training and compare it with $T$ (maybe it dominates $0.3T$).
> >
> > I would suggest for the final version to remove the part with $2^j$ (if I am not wrong and it is incorrect global normalization of the graph, please correct me here if you think opposite). For sure, mention that there should be normalization but empirically in both cases with / without normalization model converges and for simplicity we can throw away normalization. In case anyone in future encounter any problems with convergence one could try to return back $0.3T$ - this should be clearly stated.
> >
> > Would be happy to discuss further normalization topic to understand better convergence/divergence behaviour.

---

> > > ### Author Response · Authors · 2021-11-19
> > > **More discussion on normalization.**
> > >
> > > Dear Reviewer pcpu,
> > >
> > > We are glad that you are satisfied with our answers and the additional experiments. And we are happy to discuss on the normalization part.
> > >
> > > 1. The probatility should be globally normalized. Remove the $2^j$ part.
> > >
> > > A: We totally agree on the global normalization. It was indeed confusing to consider each $j$ separately (though note that, ending at j is not a single path but still an aggregation of paths). Switching to a wild-card ending state view, is much more clear. So we took this suggestion and removed the confusing $2^j$ part now.
> > >
> > > 2. Is there any normalization done on durations?
> > >
> > > A: We double-checked the code (also available in supplementary), and confirmed there is no such duration normalization done in the WCTC code. Although we use the mean loss over the batch size, this is not on the durations. The pytorch CTCLoss code does not have normalization either, but that is because original CTC has proper probability (sumed prob = 1), for every frame.
> > >
> > > 3. Why it works without normalization?
> > >
> > > A: Our theory is that, normalization here is essentially, equivalent to enforcing the same weight on each data sample. Without normalization, longer sequneces get smaller loss values. However, note that longer sequences may not be harder to learn, if they are clean audios / images. Therefore, although we didn't properly add weights to harder samples (like bootstraping), we didn't do any harms either. It just treats shorter samples more seriously than longer sequences. Short sequences could be hard as well. Therefore, the normalization part becomes less important. This is totally different than the sequence generating task, where the model tends to generate very short sequence without normalization. In that generating task, normalization is very important, but in our task, it has less influence.
> > >
> > > 4. The training loss values. Is 0.3T hidden in large loss values?
> > >
> > > A: This is a very good point. We added an additional plot in Appendix D. 9 (bottom of page 21). Here we draw the loss curves during training, of both normalized and unnormalized version. There is a constant shift due to the 0.3T term. This shift value (around 150) is larger than the unnormalized loss value (around 50), so $0.3T$ is definitely a significant part in the total loss, which is not ignorable. But the training curves for both cases are very similar, indicating that the normalization does not affect the training trend in general.
> > >
> > > It is an interesting topic worth a better understanding. Thank you for the discussions!

---

> > > > ### Comment · Reviewer_pcpu · 2021-11-21
> > > > **More, more discussion on normalization!**
> > > >
> > > > Dear authors,
> > > >
> > > > Thanks for double checking on implementation details of CTC in PyTorch. Also Appendix C now is more readable (from my point of view), and additional plots on loss scales are very helpful in having idea of the overall behaviour, thanks!
> > > >
> > > > From my experience of CTC modification I met the problem of divergence in case of no proper normalization for the graph. It is interesting to see that in your case it is still working. Hope, in future you could investigate more this effect, maybe we could discover even other properties which influence convergence/divergence. One potential thing to have a look in future work is to check dataset which has high variability in input size but balanced and evaluate on different durations in validation data separately. Compared with CTC this could give a hint on difference with it and on necessity of normalization.
> > > >
> > > > Thanks again for all additional experiments and discussions!

---

> > > > > ### Author Response · Authors · 2021-11-21
> > > > > **Thank you for the suggestions**
> > > > >
> > > > > Dear Reviewer,
> > > > >
> > > > > It is our pleasure to have such a discussion on the normalization topic. Thank you for your valueable input and concrete suggestions on checking the potential issues, as well as the directions in future work. We appreciate your active participation during the rebuttal period !

---

> > > > > > ### Comment · Reviewer_pcpu · 2021-11-27
> > > > > > **Recommendation for acceptance**
> > > > > >
> > > > > > Thanks again authors for additional experiments (including resolving my concern on wav2vec features) and detailed explanations!
> > > > > >
> > > > > > As I said before proposed idea is mainly based on existing DTW algorithm (SPRING), but important adjustments are done for CTC case. I recommend the paper for acceptance, but still prefer to stay with the original score "6: marginally above the acceptance threshold" as we still have no clear understanding what is happening with proper normalization of the graph (what is the reason why non-normalized version converge finely and has the same loss profile as normalized version).

---

### Official Review · Reviewer_VmYt · 2021-11-02

**Correctness:** 4
**Technical Novelty And Significance:** 2
**Empirical Novelty And Significance:** 3
**Recommendation:** 6
**Confidence:** 4

**Main Review:**

The problem addressed by this paper is interesting and relevant. The paper is well written overall, easy to understand with clear definitions and examples. The equations look ok and the paper explains well the complexity and how to improve the loss in the presence of partial transcripts.

The "Key summary" section is quite helpful to summarize and clarify the contributions and helps a lot to understand the whole paper. In particular, it clarifies some parts that may not be so clear for the reader in the abstract (e.g. "partial" in this work only refers to missing labels at the beginning or end, or that the proposed "wildcards" are only used for the beginning part).

The experiments on three tasks look sound and good, the method is well detailed. The ablation study on the weighting of the losses for different sub-segments is interesting.

The general definitions about alignments in the introduction and the different mentions of attention-based methods in the paper do not bring much to the paper and could be at least reduced. Attention-based methods are not addressed at all by the paper anyway. Reducing these parts would leave some room for further analysis in the experiments section.

Regarding the mask ratio in the experiments:
  - it is not clear how the masking is done for training, whether the mask is evenly distributed or not between beginning and end
  - in the figures it is not clear whether the mask ratio is the one used for training, if there is masking at inference too, and in that case, whether the mask ratio is the same for training and inference.
  - in general it would be helpful to see the results when the mask ratio is random and a comparison with CTC trained on full vs masked vs partial transcript corresponding to knowing the mask

Regarding the wildcards, why not either use wildcards at the beginning AND at the end, or apply the same trick for the beginning as the one applied at the end (i.e. allow the alignment to begin anywhere). As the authors state themselves, the fact that $p(\*|X)$ is always $1$ does not allow to formulate the whole loss as a probability. A "correct" formulation of the problem from a mathematical standpoint would have been much more interesting. For example, how about $p(\*|X) = r$ and use $(1 - p(\*|X))$ to "enter" the main graph? How about doing something similar for the end?

It is not completely clear either whether the wild-cards are used at inference as well or not.

The rationale behind weithing scheme is not completely clear. The summarized probabilities correspond to products of probabilities with a different number of factors: how to make sure that shorter segments do not have a lower loss just because less probabilities are multiplied. It would make sense to see the duration of the segment appear somewhere. Moreover, one would expect to treat the missing beginnings as the missing ends in the weighting scheme. As mentioned earlier, $\mathcal{L}^{(j)}_{CTC}$ does not correspond to a log-probability.

The CTC loss also sometimes need time to start converging, since the underlying network has to learn to align. The proposed method probably makes this issue bigger, and it would have been nice to at least see convergence curves, or an analysis of the training dynamics.

Finally, in the semi-supervised part about helping to generate actual labels: it is not exactly clear what or how samples would be discarded. Moreover, the proposed method could be used to simply do a forced alignment of the training set with a pretrained model to "clean" the dataset. One would expect to see i the paper the advantages of the proposed integrated training method vs this simple approach.

Edit after rebuttal:
The authors addressed the doubts and questions raised in the review.

**Summary Of The Paper:**

This paper presents a modification of the CTC training loss to cope with incomplete transcripts in the training set, where the actual transcription of the beginning or of the end is missing. The authors propose to minimize the loss over all possible sub-segments of the input to automatically align the one that matches the available transcript. The main contribution is that it allows to directly train the neural network with CTC without having to clean or align the data first when some transcript are only partial.

**Summary Of The Review:**

The problem is interesting and the paper is clear and easy to understand. However, the contribution of how to use dynamic programming to compute the CTC loss over all sub-segments is marginal. The paper could benefit from a deeper analysis and a more mathematically sound definition.

Edit:
The additional experiments and details are helpful and indeed benefit the paper.

---

> ### Author Response · Authors · 2021-11-15
> **Thank you for your valuable feedback and comments.**
>
> Thank you for your valuable feedback and comments. We would like to answer your questions here. We put all additional experiments/discussions during response period, in a big Appendix D in the paper, for easy to look up. In the final version we will merge that into different parts of the paper.
>
> 1. Mentioning attention models is not necessary.
>
> A: Agreed. We will remove this part to save space. Thanks for the suggestion.
>
> 2. How the masking is done?
>
> A: We applogize for not making this clear in the paper. Let us use an example to illustrate. Suppose we have a label string "helloworld", 10 characters in total. We want to mask 50%, which means 5 characters. We only mask front and end randomly. So we generate a random starting position, uniformly from [0, 5]. Suppose we get 3 from rand function, then we keep the characters from 3 to 7, and throw the rest. In this case, the remaining label for training will be "llowo". We updated the clarification in Appendix D.1.
>
> 3. Is masking the same in training and inference?
>
> A: Sorry for the confusion. Masking only applies in training, as the goal is to let the model learn from corrupted labels. It is hard to find a public dataset that contains incomplete labels (they are mostly very clean), so we apply mask in the training phase to mimic the corrupted label cases. The inference is just to test whether the model has learned the representations, so there is no masking. Hope this is clear and we have updated this clarification in Appendix D.1.
>
> 4. In general it would be helpful to see the results when the mask ratio is random and a comparison with CTC trained on full vs masked vs partial transcript corresponding to knowing the mask?
>
> A: There is a misunderstanding here. We are happy to clarify: The mask-ratio is not a parameter feed into the model. The model does not know the mask-ratio, or even whether the label is complete or masked. We set different mask-ratio just for illustrating how the model performs under lighter or heavier label corruptions. Certainly we can use a random mask-ratio. That randomness is not likely to hurt or benefit the model, as the model does not know it (strictly speaking, some data batches get worse training, but some batches get better, due to such randomness). We are glad to add such experiment in Appendix D.2. Since it is uniformly random between 0 and 1, the expectation is 0.5. So we compare with mask-ratio = 0.5, and no-mask in Table 5. As expected, the random-ratio yields similar performance as r=0.5.
>
> 5. Why not using wild-card at both sides?
>
> A: Prepending a wild-card (in front of the label) allows flexibility to start from a middle audio frame. However, appending a wild-card (in the back) has two main issues. We provide a detailed explanation in Appendix D.6. In short: (1) It is equivalent to the 'sum-prob' case in Equation 5. So it loses the flexibility to use "max-prob" or "softmax-prob" (the weighted-sum), if we want to select the best ending position. (2) In our unnormalized version, this scheme will accumulate probabilities from wild-card, and override the actual paths' probability. Like gradient vanishing, here the path probability becomes relatively too small, causing the training failure. We observed that it does not converge if wild-card at both sides. We can use the normalized version to mitigate, but it becomes much more complex than just using weighted-sum at the end.
>
> 6. Why not allowing weight-sum trick at both sides?
>
> A: It is not straight-forward to apply weight-sum in front, as this is to obtain the aggregated paths' probability. There is no path in the begining. I guess you might mean we can choose the starting position and somehow combine them in a weighted way. But the weights still needs to be computed at the end of the path. In contrast, wild-card is both conceptually simpler and easier for implementation, to handle the front part. We updated this discussion in Appendix D.6.
>
> ---> continued

---

> > ### Author Response · Authors · 2021-11-15
> > **---> continued**
> >
> > 7. A proper probability formulation of P(\*|X), e.g. making P(\*|X) = p < 1, and other characters have a sumed prob 1-p.
> >
> > A: This is a very good point. The obvious benefit using this form is the probability of each frame, can be normalized. The disadvantage is that we need to introduce another hyper-parameter p, to control the probability of getting the wild-card. This is more complicated and slightly altered the meaning of wild-card (now it is treated like a regular symbol, with an emission probability). But this also brings the potential such as dynamically adjusting this p, for different frames. We conducted an additional experiment in Appendix D.4. Please see the details there. Unfortunately, we didn't notice a big difference using this scheme. Combined with Appendix C, it is reasonable to move with the simpler unnormalized version. But we think this is still a valid point, and it could benefit if we let the model to learn the p and dynamically adjust it.
> >
> > 8. Is wild-card used in inference?
> >
> > A: No. It is only involved in training phase, to handle possible missing labels. Inference is to test whether the model successfully trained. Wild-card is not in the decoded outputs.
> >
> > 9. Rationale behind weighing scheme?
> >
> > A: Sorry for the confusion. Idealy we would like to choose the ending point with the maximum aggregated path probability. However, empirically we found that taking the hard maximum, leading to convergence issues. Therefore, a relaxed version, the soft-max (the weighted-sum) is adopted here. Our ablation study in Section 3.4 also verifies this choice.
> >
> > 10. Since the probability is not normalized, the duratioin effect needs to be considered.
> >
> > A: Yes we strongly agreed that the duration could affect the results (longer sequences get higher probability, therefore smaller loss). That's why we provide Appendix C to study this issue. Empirically we found normalization has little impact. We also added the statistics of durations in Appendix D.5. For quite a large range of duration, the method performs constantly. We have some thoughts as follows: Although in a seq2seq generative model, without normalization, the model tends to generate short sentences, which is unwanted. However, here we are dealing with a different problem, the higher prob essentially reduce some weights to longer utterances. The model is encouraged to pay more attention to the correctness of shorter sequences. But this does not necessarily lead to poor performance. Shorter audios could be either clear (easy to learn) or noisy (hard to learn). It is like bootstraping, but in a random fashion. So it does not bring gains, neither hurts. Empirically we also didn't observe difference whether normalize or not (Appendix C and D.4). So we choose the unnormalized version due to simplicity. Again, this is not theoretically proved. We add this discussion in Appendix D.5.
> >
> > 11. Convergence behaviors.
> >
> > A: Thanks for this suggestion and we added the convergence curves in Appendix D.3. As it shows, the convergence of WCTC is the same as standard CTC, when label is clean, and much better than standard CTC when label is incomplete. The detailed discussion can be found at Appendix D.3. Hope this part addresses your concerns.
> >
> > 12. Discussion on the semi-supervised learning example is not clear.
> >
> > A: Sorry for the confusion. This is a motivating example. We want to say that, for example, have a recognizer trained on one dataset, and we can use it to generate pseudo-labels on another dataset. The recognizer also outputs confidence for each character in the pseudo label. We can discard the characters with low confidence, and use the remaining incomplete labels to train the target recognizer. Partial but correct labels, would be better than full-but-wrong labels.
> >
> > 13. Comparison with force-alignment methods.
> >
> > A: This is a good point. If we have good force aligner, the force-alignment could be done beforehand. However, as also pointed by reviewer sfbX, we often face the issue in the segmentation, when we don't have a good force-aligner. This is particularly true when processing videos / images. With the proposed method, we can avoid accurate segmentation and simply leave some redundant part in both sides, then we can still make the training possible without significant performance drops.
> >
> > Thank you again for all the valuable comments and suggestions. We answered the questions and provide additional materials as above. We hope these address your concerns and please take them into consideration for the final scores. We are happy to answer additional questions.

---

### Official Review · Reviewer_sfbX · 2021-11-03

**Correctness:** 3
**Technical Novelty And Significance:** 3
**Empirical Novelty And Significance:** 3
**Recommendation:** 6
**Confidence:** 4

**Main Review:**

strengths:
- The missing label issues often happen in the read training data.
- I often faced this kind of issue when we try to use long recordings (e.g., youtube, ted corpus, or podcast data). In this case, the segmentation time stamp is not accurate and the missing labels or missing audios in the edge often happen. We usually use some force-alignment techniques to realign such data, but this method can be applied to such data directly. (Note that the missing audio issues can be solved by just using longer segmentation that would be supposed to cover the contents corresponding to the transcriptions).
- CTC is now widely used as an alternative seq2seq model (originally only ASR/OCR, but now NMT and speech translation). This method would have a broad impact on the machine learning community.
- The experimental effectiveness is valid. It shows robust learning behaviors. Also, it was shown in three different tasks.

weaknesses
- The experiments are not real. I recommend the authors try some long recording setup as I mentioned before.


**Summary Of The Paper:**

This paper proposes an extension of CTC by considering the wild-card to adjust the label missing issues during training, which tends to happen in the onset/offset edges of the utterance. The paper elegantly formulates it as an extension of the CTC framework and applies it to two tasks (ASR and OCR). The method shows robust training behaviors compared with the original CTC training.

Other comments:
- Can we use it for RNN transducer?
- I'm expecting that this method may work even if missing labels happen in the middle if we use the self-attention-based network. The self-attention network can handle some re-ordering (move the hidden vectors corresponding missing part to the edge).
- Section 1.1, the definition parts. These parts are well written, but the discussion about <blank> is missing.
- I mentioned below, but this problem can be applied to the missing audio (missing X) case in the long recording scenario by just using a longer segment to cover all labels. So, for my major applications of this method, I don't think that this becomes a limitation.
- does the wild-card symbol contain the <blank> symbol? It may not be a matter, but I'm curious.
- It would be better if the paper has more discussions of why 'max-prob' does not work and 'weighted-sum" works the best.
- Is it possible to correctly find the wild card region with the Viterbi algorithm? I think this is a very good option. I also want to verify that the wild card region is correctly identified (I think Figure 4 shows it to some extend, but I want to know more analysis/examples).




**Summary Of The Review:**

This paper would have a broad impact on the machine learning community to tackle noisy data training based on CTC. The formulation is straightforward and the experimental effectiveness is valid. The paper does not show the experimental impact of this method on the real problem, and this addition makes the paper stronger.

---

> ### Author Response · Authors · 2021-11-16
> **Thank you so much for your positive feedback!**
>
> Thank you so much for your positive feedback! We would like to answer your questions as follows. Meanwhile, we updated our paper with a big Appendix D, to collect all additional experiments/discussions. This is easy to look up. In the final version we will merge it into the paper.
>
> 1. Can this be used in RNN-Transducer?
>
> A: Yes it can! Since RNN-Transducer training also builds the trellis, uses dynamic programing to obtain and maximize the likelihood of gold labels, this part is the same as CTC training. So we can apply the proposed method here as well.
>
> 2. Can we use it in self-attention models, where missing label is in the middle and attention handles the reordering?
>
> A: This is an interesting idea, as more and more works on joint CTC and attention training are coming up. So far, none of them tries to solve the missing label problem. We highly appreciate this idea. This would require non-trivial works so we leave it for future work.
>
> 3. Lack of discussion of <blank> in section 1.1 definition part.
>
> A: Sorry for the missing definition. <blank> was brought up with the original CTC paper [graves2006], so we simply mention it without proper definition. We will update this part.
>
> 4. Missing X cases seems not a problem as we can record long audio/videos to cover all labels.
>
> A: Agreed. In practice, indeed we can avoid it by recording longer audio/videos. We might still face missing X problem when data is already collected and hard to make any adjustment. We updated the paper's limitation part accordingly.
>
> 5. Does wild-card contain <blank> ?
>
> A: Yes it matches any character so it contains <blank>.
>
> 6. Why max-prob does not work, but weighted-sum works ?
>
> A: This is mostly an empirical choice. Intuitively speaking, the hard choice (take the max one), will ignore the frames beyond that ending point. For example, if we have 10 frames, and the path ending at the 8th frame, got the maximum probability. Once we use max-prob, we will completely ignore the 9th and 10th frames. As a result, the gradient will not back-propogate to these 2 frames. The soft version (weighted-sum) mitigate this problem to some extent. Again, this is not therotically justified, but just our suspect. We have added this discussion in Appendix D.6.
>
> 7. Find wild-card region with Viterbi algorithm ?
>
> A: We are not sure if this applies. Viterbi algorithm is used to select the emissions given a hidden Markov model, using dynamic programming. If wild-card is one of the emission, and transition probability from wild-card to wild-card is 1, then it will always output wild-cards. But here in W-CTC, wild-card is not one of the emissions (the ASR model does not produce wild-card symbol). The trellis is different from Viterbi's trellis conceptually. The wild-card here only alter the path, and the path still needs to conver all label symbols. Therefore, an all-wildcard paths is not allowed. Hope this answers your question.
>
> 8. More examples to show the wild-card region and analysis.
>
> A: We added more example images in Appendix D.8. In particular, we grouped the examples in three categories to compare standard CTC with W-CTC. We have a new finding that, even trained with clean label, the standard CTC still fails to find the correct alignment path if the missing part is in front. In the original "beautifully" example, since the remaining label is "eaut", very close to the front, the standard CTC can identify a reasonable path, though with low confidence. But in other cases, the standard CTC fails. We also show some examples that having confusing paths, e.g., the corrupted label has overlap to more than 1 positions in the image. In all the cases, we can see that the proposed W-CTC is able to find correct path and enable models to learn from that. Thanks for this suggestion and please refer to Appendix D.8 for more details.
>
> 9. Experiments on real dataset with long-recordings.
>
> A: In fact, this paper is motivated when we tried to preprocess (segment) some collected data. But for paper writing, we have to run on public datasets, which are mostly clean. It is a bit difficult to find public dataset with incomplete labels for training, where testing label is full. The data itself is not synthetic but lable is randomly masked. We strongly agree that the method would help when accurate segmentation is hard to get. We thank the reviewer pointing this out and appreciate the suggestions to handle the missing audios.
>
> We thank the reviewer again for all the insightful comments and hope to address all the concerns.

---

### Comment · Area_Chair_wuU7 · 2021-11-15
**Please update your ratings if needed based on the authors' responses**

Dear Reviewers,

The authors have made detailed responses to all the reviews. Please take a look and see whether they address your concerns and update the ratings if necessary. Thanks for your help and expertise!

---

### Author Response · Authors · 2021-11-16
**We have updated the paper with additional experiments/analysis.**

Dear reviewers,

We thank for your reviews and highly appreciate all the valuable comments and suggestions. We have answered the raised questions and updated our paper with one big Appendix D, from page 14 to page 21, to include all the additional experiments, analysis and discussions. This is easy to look up during the response peroid. We will merge it into different parts of the paper in the final version.

Please kindly review the answers and the paper updates. We hope these address your concerns and hope the updates can be taken into consideration for the scores. We are happy for further discussions.

Thank you!

---

### Comment · Area_Chair_wuU7 · 2021-11-24
**Reminder to adjust the ratings if needed**

Dear Reviewers,

The authors have made detailed responses to the reviews and revised the paper. Please take a look and see whether they address your concerns and update the ratings if necessary. Thanks for your help and expertise!

---

### Decision · Program_Chairs · 2022-01-20

**Decision:**

Accept (Poster)

**Comment:**

This paper proposes an extension of CTC by considering the wild-card to adjust the label missing issues during training. The authors propose to minimize the loss over all possible sub-segments of the input to automatically align the one that matches the available transcript. It is empirically proved to significantly improve performance over CTC even if up to 40-70% label sequence is missed (overall performance similar to the complete label case) across different tasks.

As agreed by the reviewers, the paper is well presented and the problem is interesting to a broad community. Dynamic time warping with unconstrained endpoints itself is not a new idea and a classical topic for speech recognition (e.g. word spotting). The contribution of the paper is the formal introduction of the approach to CTC and to give experimental results to confirm the effectiveness. Also the use of simulated data weakens the paper a bit.

The decision is mainly based on the clear presentation and fair experimental justification.